# RNA polymerase II primes Polycomb-repressed developmental genes throughout terminal neuronal differentiation

Carmelo Ferrai[1,2,3,*,†] iD, Elena Torlai Triglia[1,†] iD, Jessica R Risner-Janiczek[3,4,5], Tiago Rito[1], Owen JL Rackham[6], Inês de Santiago[2,3,§], Alexander Kukalev[1], Mario Nicodemi[7], Altuna Akalin[8], Meng Li[3,4,¶], Mark A Ungless[3,5,**] iD & Ana Pombo[1,2,3,9,***] iD

## Abstract

Polycomb repression in mouse embryonic stem cells (ESCs) is tightly associated with promoter co-occupancy of RNA polymerase II (RNAPII) which is thought to prime genes for activation during early development. However, it is unknown whether RNAPII poising is a general feature of Polycomb repression, or is lost during differentiation. Here, we map the genome-wide occupancy of RNAPII and Polycomb from pluripotent ESCs to non-dividing functional dopaminergic neurons. We find that poised RNAPII complexes are ubiquitously present at Polycomb-repressed genes at all stages of neuronal differentiation. We observe both loss and acquisition of RNAPII and Polycomb at specific groups of genes reflecting their silencing or activation. Strikingly, RNAPII remains poised at transcription factor genes which are silenced in neurons through Polycomb repression, and have major roles in specifying other, non-neuronal lineages. We conclude that RNAPII poising is intrinsically associated with Polycomb repression throughout differentiation. Our work suggests that the tight interplay between RNAPII poising and Polycomb repression not only instructs promoter state transitions, but also may enable promoter plasticity in differentiated cells.

**Keywords** cell plasticity; chromatin bivalency; gene regulation; RNA polymerase II; transcriptional poising
**Subject Categories** Chromatin, Epigenetics, Genomics & Functional Genomics; Genome-Scale & Integrative Biology; Transcription

**Mol Syst Biol.** (2017) 13: 946

## Introduction

Embryonic differentiation starts from a totipotent cell and culminates with the production of highly specialized cells. In ESCs, many genes important for early development are repressed in a state that is poised for subsequent activation (Azuara *et al*, 2006; Bernstein *et al*, 2006; Stock *et al*, 2007; Brookes *et al*, 2012). These genes are mostly GC-rich (Deaton & Bird, 2011), and their silencing in pluripotent cells is mediated by Polycomb repressive complexes (PRCs). Genes with more specialized cell type-specific functions are neither active nor Polycomb repressed in ESCs, have AT-rich promoter sequences, and their activation is associated with specific transcription factors (Sandelin *et al*, 2007; Brookes *et al*, 2012).

Polycomb repressive complex proteins have major roles in modulating gene expression during differentiation and in disease (Prezioso & Orlando, 2011; Richly *et al*, 2011). They assemble in two major complexes, PRC1 and PRC2, which catalyze H2AK119 monoubiquitination and H3K27 methylation, respectively (Simon & Kingston, 2013). Both PRC-mediated histone marks are important for chromatin repression, and synergize in a tight feedback loop to recruit each other's modifying enzymes (Blackledge *et al*, 2015).

1 Epigenetic Regulation and Chromatin Architecture, Max Delbrück Center for Molecular Medicine, Berlin, Germany
2 Genome Function, MRC London Institute of Medical Sciences (previously MRC Clinical Sciences Centre), London, UK
3 Institute of Clinical Sciences (ICS), Faculty of Medicine, Imperial College London, London, UK
4 Stem Cell Neurogenesis, MRC London Institute of Medical Sciences (previously MRC Clinical Sciences Centre), London, UK
5 Neurophysiology Group, MRC London Institute of Medical Sciences (previously MRC Clinical Sciences Centre), London, UK
6 Duke-NUS Medical School, Singapore, Singapore
7 Dipartimento di Fisica, Università di Napoli Federico II and INFN Napoli, Complesso Universitario di Monte Sant'Angelo, Naples, Italy
8 Scientific Bioinformatics Platform, Berlin Institute for Medical Systems Biology, Max Delbrück Center for Molecular Medicine, Berlin, Germany
9 Institute for Biology, Humboldt-Universität zu Berlin, Berlin, Germany
*Corresponding author. Tel: +49 3094061755; E-mail: carmelo.ferrai@mdc-berlin.de
**Corresponding author. Tel: +44 2083838299; E-mail: mark.ungless@imperial.ac.uk
***Corresponding author (lead contact). Tel: +49 3094061760; E-mail: ana.pombo@mdc-berlin.de
†These authors contributed equally to this work
§Present address: Seven Bridges Genomics UK Ltd, London, UK
¶Present address: Neuroscience and Mental Health Research Institute, School of Medicine and School of Biosciences, Cardiff, UK

Although imaging studies suggest that PRC-repressed chromatin has a compact conformation (Francis *et al*, 2004; Eskeland *et al*, 2010; Boettiger *et al*, 2016), molecular and cell biology approaches show that PRC repression in ESCs coincides with the occupancy of poised RNAPII complexes and active histone marks in the vast majority of PRC-repressed promoters (Azuara *et al*, 2006; Bernstein *et al*, 2006; Stock *et al*, 2007; Brookes *et al*, 2012; Tee *et al*, 2014; Kinkley *et al*, 2016; Weiner *et al*, 2016). The co-occurrence of RNAPII and PRC enzymatic activities, RING1B and EZH2, on chromatin has been confirmed in ESCs by sequential ChIP (Brookes *et al*, 2012) and is mirrored by the simultaneous presence of H3K4me3 and H3K27me3 marks, called chromatin bivalency (Azuara *et al*, 2006; Bernstein *et al*, 2006; Voigt *et al*, 2012; Kinkley *et al*, 2016; Sen *et al*, 2016; Weiner *et al*, 2016).

RNAPII function is regulated through complex post-translational modifications at the C-terminal domain (CTD) of its largest subunit, RPB1, which coordinate the co-transcriptional recruitment of chromatin modifiers and RNA processing machinery to chromatin, leading to productive transcription events and mRNA expression (Brookes & Pombo, 2009; Zaborowska *et al*, 2016). In mammals, the CTD comprises 52 repeats of the heptapeptide sequence N-Tyr1-Ser2-Pro3-Thr4-Ser5-Pro6-Ser7-C. At active genes, RNAPII is phosphorylated on Ser5 (S5p) to mark transcription initiation, on Ser7 (S7p) during the transition to productive transcription, and on Ser2 (S2p) during elongation. S5p and S7p are mediated by CDK7, while S2p is mediated by CDK9. RNAPII also exists in a paused state of activation characterized by short transcription events at promoter regions, followed by promoter-proximal termination and re-initiation events (Adelman & Lis, 2012). RNAPII pausing is identified at genes that produce mRNA at lower levels and is often measured as the amount of RNAPII at gene promoters relative to its occupancy throughout coding regions. Paused states of RNAPII are therefore a feature of genes that are active to a lower extent, are characterized by the presence of S5p and S7p at gene promoters, low abundance of S2p throughout the coding regions, and they are recognized by 8WG16, an antibody which has a preference for unphosphorylated Ser2 residues. The paused RNAPII complex is also characterized by methylation and acetylation of the non-canonical Lys7 residues at the CTD (Schröder *et al*, 2013; Dias *et al*, 2015; Voss *et al*, 2015).

The RNAPII complex that primes PRC-repressed genes in mouse ESCs has a unique configuration of post-translational modifications of the CTD, which is different from the paused RNAPII, and was originally referred to as *poised* RNAPII (Stock *et al*, 2007; Brookes & Pombo, 2009). The poised RNAPII is characterized by exclusive phosphorylation of S5p in the absence of S7p, S2p, K7me1/2, K7ac, or recognition by 8WG16 (Brookes *et al*, 2012; Dias *et al*, 2015). Poised RNAPII-S5p, in the absence of 8WG16, has not been described in *Drosophila* (Gaertner *et al*, 2012), consistent with lack of chromatin bivalency (Vastenhouw & Schier, 2012; Voigt *et al*, 2013). Importantly, Ser5 phosphorylation of poised RNAPII complexes at Polycomb-repressed genes in ESCs is mediated by different kinases, ERK1/2 (Tee *et al*, 2014; Ma *et al*, 2016), instead of CDK7 which phosphorylates both S5p and S7p at active genes, irrespectively of pausing ratio (Akhtar *et al*, 2009; Glover-Cutter *et al*, 2009). Loss of ERK1/2 activity in ESCs results in the loss of poised RNAPII-S5p and decreased occupancy of PRC2 at

Polycomb-repressed developmental genes (Tee *et al*, 2014), suggesting a tight functional link between the presence of poised RNAPII-S5p at Polycomb-target genes and the recruitment of Polycomb.

While histone bivalency has been studied to some extent during mammalian cell differentiation and found present at smaller proportion of genes (Mohn *et al*, 2008; Lien *et al*, 2011; Wamstad *et al*, 2012; Xie *et al*, 2013), it remains unexplored whether the co-occupancy of poised RNAPII-S5p at PRC targets is a property of ESCs or extends beyond pluripotency. The poised RNAPII-S5p state was observed at Polycomb-repressed genes in ESCs grown in the presence of serum and leukemia inhibitor factor (LIF; Stock *et al*, 2007; Brookes *et al*, 2012; Tee *et al*, 2014; Ma *et al*, 2016). Other studies grow ESCs in 2i conditions to simulate a more naïve pluripotent state, through inhibition of GSK3 and MEK signaling, which in turn inhibits ERK signaling. In these conditions, the occupancy of poised RNAPII complexes is reduced at Polycomb-target genes (Marks *et al*, 2012; Williams *et al*, 2015), consistent with the effects of ERK1-2 inhibition (Tee *et al*, 2014). Interestingly, the decreased occupancy of poised RNAPII-S5p at PRC-repressed genes in 2i conditions is accompanied by reduced occupancy of PRC2 catalytic subunit EZH2 and H3K27me3 modification, suggesting a tight interplay between the presence of poised RNAPII-S5p and Polycomb occupancy at Polycomb-repressed genes in ESCs, which is interfered upon in 2i conditions. Interestingly, prolonged 2i treatment was shown to impair ESC developmental potential and cause widespread loss of DNA methylation (Choi *et al*, 2017; Yagi *et al*, 2017), leading to renewed interest in understanding the regulation of developmental genes, and in particular whether poised RNAPII complexes are a more general feature of Polycomb repression mechanisms in the early and late stages of differentiation. Recent studies of DNA methylation in differentiated tissues show that many silent developmental regulator genes remain hypomethylated in wide genomic regions (also called DNA methylation valleys; or DMVs) in differentiated tissues (Xie *et al*, 2013), raising the question of whether poised RNAPII complexes might prime developmental regulator genes in cell lineages irrespectively of future activation. In this scenario, Polycomb repression may represent the universal mode of repression of this group of CpG-rich genes that recruit poised RNAPII-S5p and which are not targeted by DNA methylation. Silencing of developmental regulator genes through Polycomb repression mechanisms in fully differentiated cells, especially in the presence of poised RNAPII complexes, may nevertheless have roles in the remodeling of cell function, for example in response to specific stimuli such as tissue injury, or in disease such as in cancers associated with Polycomb dysfunction.

To investigate RNAPII poising at Polycomb-repressed genes, from pluripotency to terminal differentiation, we mapped H3K27me3 (a marker of Polycomb repression), RNAPII-S5p (present at both active and poised RNAPII complexes), and RNAPII-S7p (a marker of productive gene expression) and produced matched mRNA-seq datasets in ESCs and in four stages of differentiation of functionally mature dopaminergic neurons. We show compelling evidence that the presence of poised RNAPII at H3K27me3-marked chromatin is not a specific feature of ESCs, but a general property common to differentiating and post-mitotic cells. We also observe *de novo* Polycomb repression during neuronal cell commitment and neuronal maturation that promotes waves of transient downregulation of gene expression. We discover a group of genes that maintain poised RNAPII-S5p and Polycomb silencing throughout neuronal

differentiation, and which are developmental transcription factors important for cell specification toward non-neuronal lineages. Although these genes are unlikely to be subsequently reactivated in the neuronal lineage, their silencing in neuronal precursors and mature neurons is sensitive to Polycomb inhibition or knockout. We also show that the presence of poised RNAPII-S5p at specific subsets of Polycomb-repressed genes in terminally differentiated neurons coincides with their wide hypomethylation in mouse brain. Our study reveals the interplay between RNAPII poising and Polycomb repression in the control of regulatory networks and cell plasticity throughout cell differentiation.

# Results

### Capturing distinct stages of differentiation from ESCs to dopaminergic neurons

To study the dynamic changes in Polycomb and RNAPII occupancy at gene promoters during differentiation, we optimized neuronal differentiation protocols to obtain large quantities of pure cell populations required for mapping chromatin-associated histone marks and RNAPII at five states of neuronal differentiation that leads to the production of functional dopaminergic neurons (ESC, days 1, 3, 16, and 30; Fig 1A). To capture the early exit from pluripotency, we adopted an approach that starts from mouse ESCs grown in serum-free and 2i-free conditions and which within 3 days achieves synchronous exit from pluripotency toward the production of neuronal progenitors (Abranches *et al*, 2009; Fig EV1A, top row). To obtain terminally differentiated dopaminergic neurons, we used an approach that commits ESCs to a midbrain neuron phenotype (Jaeger *et al*, 2011; Fig EV1A, bottom row).

The expression of pluripotency markers *Nanog* and *Oct4* decreases dramatically at days 1 and 3 of differentiation, respectively (Fig 1B). The early differentiation marker *Fgf5* is transiently expressed in days 1–3, whereas neuronal markers *Blbp, Hes5,* and *Mash1/Ascl1* are increasingly expressed from day 2 (Figs 1B and EV1B). The expression of *Sox2*, which encodes for a transcription factor expressed in ESCs and by most central nervous system progenitors (Graham *et al*, 2003), is detected from ESC to day 4, as expected (Fig EV1B; Abranches *et al*, 2009). After sixteen days, we obtained neurons that no longer express OCT4 protein, are positive for the neuronal marker TUBB3 (detected by Tuj1 antibody), and no longer divide, as assessed by lack of BrdU incorporation into newly replicated DNA (Fig 1C).

To confirm the dopaminergic phenotype, we performed immunofluorescence for tyrosine hydroxylase (TH), the rate-limiting enzyme in dopamine synthesis (Fig 1D). The neuronal populations obtained expressed TH from day 16, reaching close to ubiquitous expression at day 30 (Fig 1D). Moreover, 70% of cells co-express LMX1A and FOXA2 at day 16 (Fig EV1C), two markers specific to the dopaminergic ventral midbrain, confirming that the neuronal populations produced are highly enriched for the dopaminergic lineage (Hegarty *et al*, 2013). Taken together, these results indicate that day 16 neurons represent an immature stage of differentiation committed to the ventral midbrain lineage, which further mature until day 30.

To characterize the changes in gene expression that accompany neuronal differentiation, we produced mRNA-seq datasets for ESCs, day 1, day 3, day 16, and day 30. We identified 4656 genes whose expression levels peaked at one specific time point (Fig 1E). Genes peaking in ESCs are enriched in Gene Ontology (GO) terms typical of pluripotency, such as *"stem cell maintenance", "regulation of gene silencing",* and *"sugar utilization"*, and include *Nanog, Tet1,* and *Hk2*. Genes with highest expression on day 1 have roles in the exit from pluripotency, such as *Wt1, Foxd3,* and *Dnmt3b,* and are enriched in GO terms *"cell morphogenesis", "pattern specification process",* and *"gene silencing"*. On day 3, the expression of *Fgf8, Gli3* and *HoxA1* peaked, reflecting an early stage of neuronal commitment, highlighted by enrichment in GO terms such as *"cellular developmental process", "multicellular organismal process",* and *"neuronal nucleus development"*. Day 16 coincided with highest expression of genes associated with GO terms such as *"nervous system development", "axon guidance",* and *"neuron migration"* (including *Nova1, Sema3f, Ascl1, Neurog2*), and day 30 with genes important for dopaminergic synaptic transmission, for the *"G-protein coupled receptor protein signaling pathway"* and *"response to alkaloid"* (such as *Lpar3, Th, Park2, Chrnb4*). The complete list of enriched GO terms is presented in Dataset EV1. These expression profiles show that each time point captures a specific stage of neuronal development, and suggest that days 16 and 30 reflect early and late stages, respectively, of maturation of dopaminergic neurons.

To further confirm the quality of our samples, we also explored the expression profiles of specific single genes (Fig EV2). In addition to confirming the expression of the differentiation markers studied by quantitative PCR and immunofluorescence (Fig EV2A), we also found that the proneural gene *Ngn2* (expressed in immature, but not in mature, dopaminergic neurons) peaks at day 16 and drops by day 30, while *Nurr1* (required for maintenance of dopaminergic neurons) is upregulated at day 16 but remains expressed at day 30 (Fig EV2B; Ang, 2006). Other markers of dopaminergic neurons,

---

**Figure 1.** **Model of differentiation from pluripotent stem cells to terminal dopaminergic neurons.**

A  Schematic representation of the differentiation system used and the temporal expression dynamics of differentiation stage markers.

B  RNA levels of differentiation stage markers were measured by qRT–PCR. Relative levels are normalized to *Actb* internal control, and values are plotted relative to the highest expressed time point. Mean and standard deviation (SD) are from three biological replicates.

C  Indirect immunofluorescence confirms expression of stage-specific markers at the single cell level. OCT4 is a marker of pluripotent ESCs. Tuj1 is an antibody that detects neuronal marker TUBB3 at day 16 and day 30 neurons. The cycling activity of ESCs, day 16, and day 30 neurons was assessed by BrdU incorporation (24 h) into replicating BrDNA. Nuclei are counterstained with DAPI. Scale bar, 100 μm.

D  Tyrosine hydroxylase (TH; in red) is a marker of dopaminergic neurons. It is not expressed in ESCs and detected weakly in day 16 and broadly in day 30 neurons. Nuclei are counterstained with DAPI. Scale bar, 100 μm.

E  Gene expression dynamics across the differentiation time line for genes whose expression peaks in a single time point (z-score > 1.75; from mRNA-seq). Representative enriched GO terms were calculated using as background all genes expressed (> 1 TPM) in at least one time point. *n*, number of genes per group. Permute *P*-value (GO-Elite) is shown.

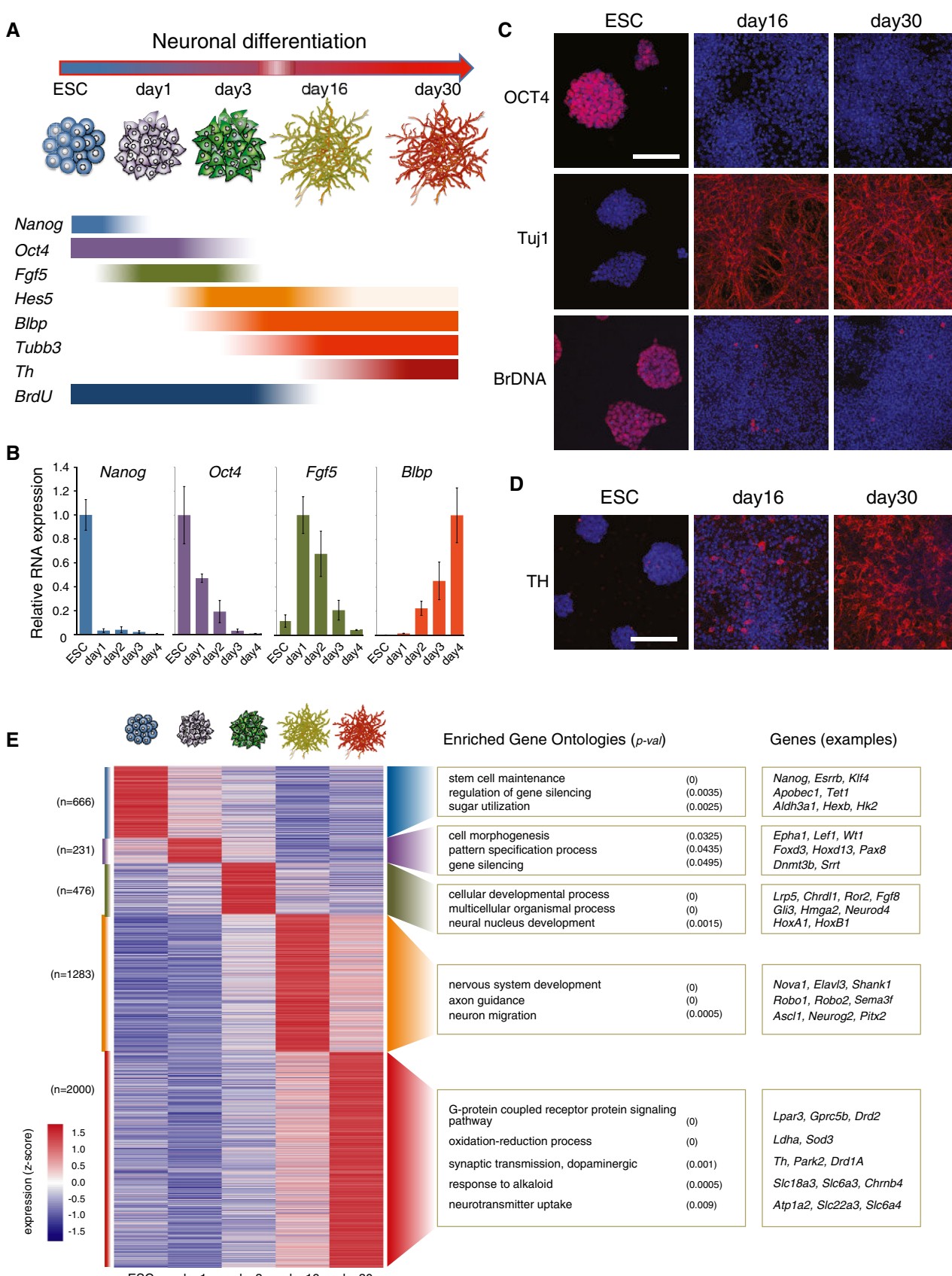

**Figure 1.**

such as *Pitx3, Aadc, Vmat,* and *Dat,* are most highly expressed at day 30 (Fig EV2B).

## Electrophysiological measurements demonstrate distinct stages of neuronal maturation on days 16 and 30 of differentiation

To directly investigate the state of maturation of neurons upon prolonged culture, we measured their action potential activity and synaptic connectivity by conducting targeted whole-cell electrophysiological recordings (Fig 2A) at four different time points (days 14–16, 20–25, 26–30, and > 30). We find that days 14–16 neurons are largely silent, whereas at days > 30 they exhibit robust spontaneous action potential activity (Figs 2B and EV3A), similar to the activity of midbrain dopaminergic neurons from *ex vivo* slice preparations and *in vivo* (Marinelli & McCutcheon, 2014). During maturation, neurons also exhibit a hyperpolarization-activated inward (Ih) current

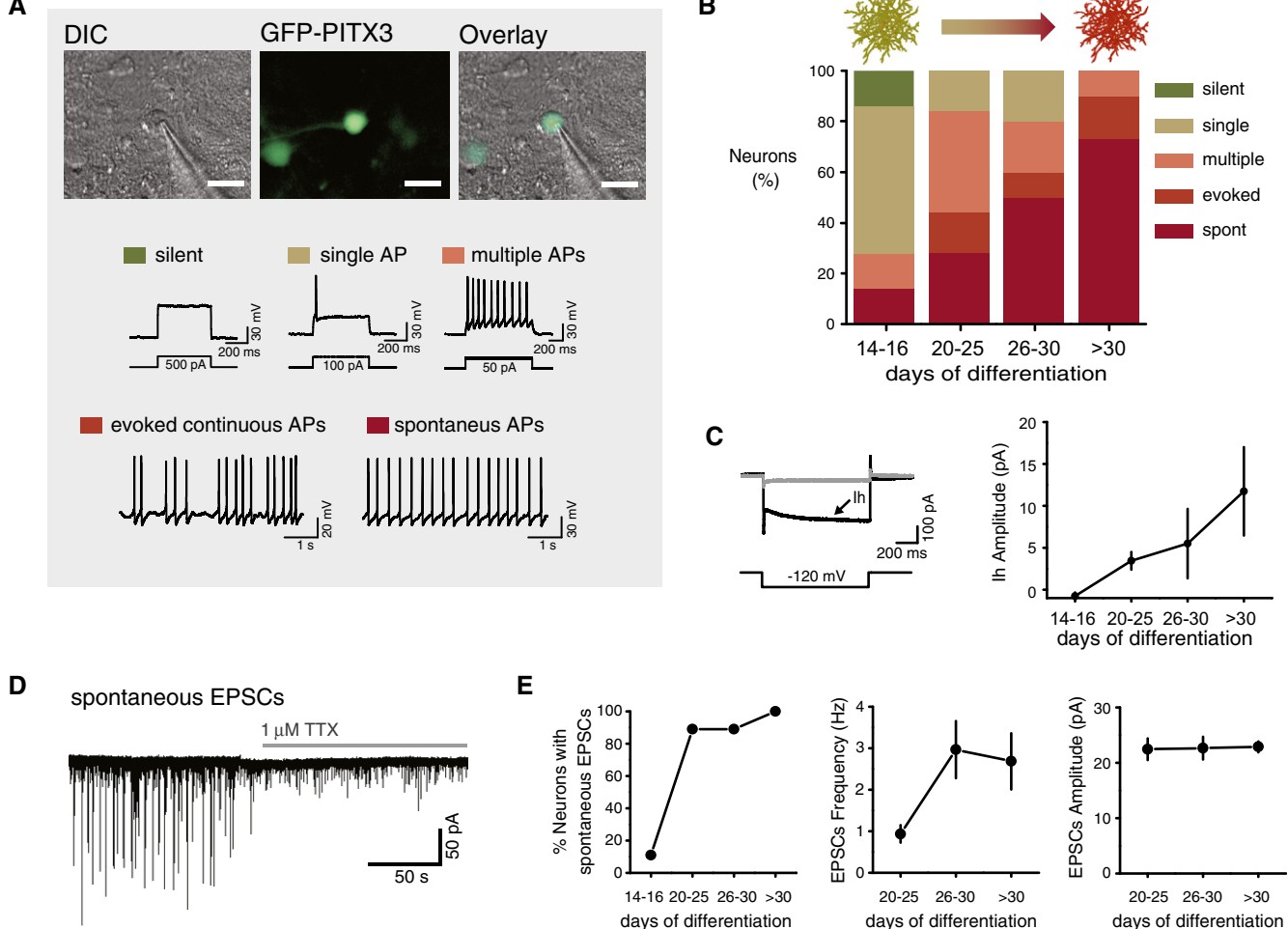

**Figure 2. Whole-cell electrophysiological recordings from *in vitro* differentiated dopaminergic neurons show maturation of action potential firing and synaptic activity.**

A   Differential Interference Contrast (DIC) and GFP images showing a glass micropipette recording from a Pitx3-GFP[+] cell and examples of action potential (AP) measurements. Scale bars, 35 μm. Immature neurons were generally silent or fired only a single AP; they progressively started firing multiple APs in response to a depolarizing current pulse, evoked spontaneous APs generated in response to a constant depolarizing current injection, or spontaneous pacemaker-like APs when they were fully mature.

B   Graph shows progressive functional neuronal maturation toward spontaneous AP firing from days 14–16 to day 30 based on the % of their AP firing categories (*n* = 21, 25, 10, 30; Spontaneous firing; chi-squared test for trend = 20.58, *P*-value < 0.0001).

C   Two traces from different cells illustrating the presence (black, arrow) or absence (gray) of a hyperpolarization-activated inward current (Ih), evoked by a hyperpolarizing voltage pulse of −120 mV. Graph shows mean and standard error of the mean (SEM). Ih amplitude progressively increases across differentiation days (*n* = 22, 10, 12, 20; ANOVA = 2.565, *P*-value = 0.0483).

D   Example trace of spontaneous excitatory post-synaptic currents (sEPSCs) recorded at −70 mV and application of 1 μM tetrodotoxin (TTX), which revealed miniature EPSCs (mEPSCs).

E   Spontaneous synaptic activity emerged at days 20–25. Graph shows percentage of neurons with spontaneous synaptic activity across differentiation days (*n* = 10, 19, 8, 5; chi-squared test for trend = 20.76, *P*-value < 0.0001). Frequency, but not amplitude of mEPSCs, continues to increase once synaptic activity has emerged (*n* = 7, 7, 10; Frequency ANOVA = 6.555, *P*-value = 0.0377; Amplitude ANOVA = 0.04457, *P*-value = 0.978). Graph shows SEM.

(Fig 2C), an electrophysiological feature commonly used to identify dopaminergic neurons (Ungless & Grace, 2012). Strikingly, after prolonged culture many cells exhibit burst-like events (Fig EV3B and C), often seen *in vivo* in midbrain dopaminergic neurons (Grace & Bunney, 1984), which are thought to be driven in part by synaptic inputs (Paladini & Roeper, 2014). We also observed maturation of functional synaptic connectivity. Spontaneous synaptic events were rare at days 14–16, but were present in most neurons from days 20–25 onwards (Fig 2D and E). Further experiments in the presence of TTX (which blocked action potential activity; Fig EV3D) revealed the presence of glutamatergic AMPA receptor-mediated miniature excitatory post-synaptic currents (mEPSCs; Fig EV3E) and GABAA receptor-mediated miniature inhibitory post-synaptic currents (mIPSCs; Fig EV3F), similar to those seen in *ex vivo* dopaminergic neurons. Consistent with this, we also observed a small number of GABAergic and glutamatergic neurons in our cultures (Fig EV3G), as seen in the ventral tegmental area (VTA) of the midbrain *in vivo*, and expression of GABAergic and glutamatergic neuronal markers (Fig EV3H). Further evidence in support of the quality of our neuronal samples was the absence of glutamatergic autapses (Fig EV3I), which are seen in cultured *ex vivo* dissociated neurons, but not in *ex vivo* brain slices (Sulzer *et al*, 1998).

We conclude that the neuronal differentiation approach established here yields functional neurons that undergo progressive maturation, as shown by their action potential activity and synaptic connectivity. Taken together, gene expression and electrophysiological analyses confirm that the five time points used here represent distinct stages of neuronal fate commitment and maturation.

## Polycomb repression dynamics during terminal differentiation of dopaminergic neurons

To identify the genes regulated by Polycomb during neuronal specification, we mapped the genome-wide occupancy of H3K27me3, the PRC2-mediated histone modification. In ESCs grown in serum-free conditions, 4,107 genes are marked by H3K27me3 (Fig 3A), most of which (78–86%) were also found targeted by PRC in ESCs in serum-containing media (Lienert *et al*, 2011; Young *et al*, 2011; Brookes *et al*, 2012), demonstrating the important roles of Polycomb repression mechanisms in ESCs independent from the use of serum.

Upon early exit from the pluripotent state, the number of H3K27me3$^+$ genes increases to ~4,600 in days 1 and 3, before decreasing to ~2,200 genes in days 16 and 30 (Fig 3A). The abundance of PRC-marked promoters in dopaminergic neurons suggests an important role of Polycomb repression in differentiated cells, which is consistent with previous reports in other specialized cell types: pyramidal glutamatergic neurons (2,178 genes; Mohn *et al*, 2008), striatal GABAergic neurons (2,057 genes; von

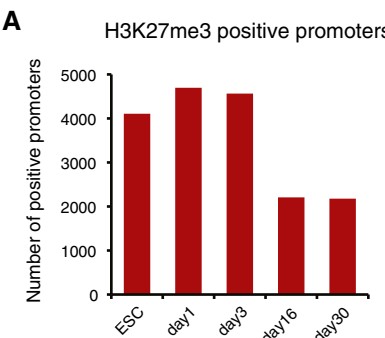

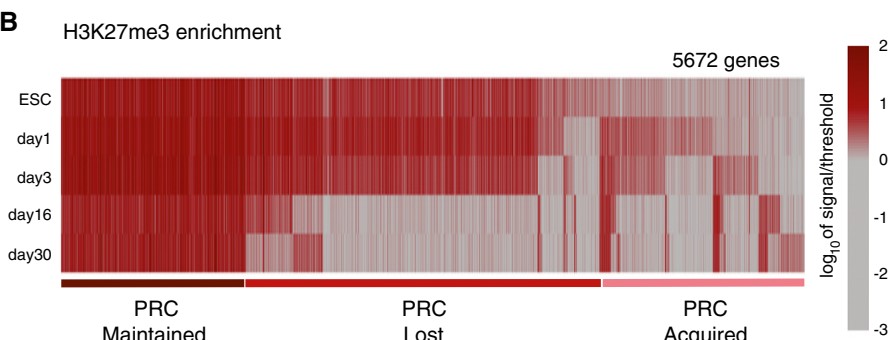

**C**

Enriched Gene Ontologies (*p-val*)

**PRC Maintained**

anatomical structure development (0)
  *Egfr, Gata4, Myod1, Nkx2-5, Pax6*
pattern specification process (0)
  *Bmp7, Foxa1, Gsc, Hoxa1, Mesp1*
regulation of cell differentiation (0)
  *Cebpa, Hmga2, Kit, Sox21, Tgfb1*
transcription, DNA-dependent (0)
  *Meis1, Hlx, Eomes, Pitx1, Twist1*

**PRC Lost**

nervous system development (0)
  *Sema3c, Igf1, Fos, Dscam, Epha4*
regulation of signaling (0)
  *Zic1, Fgfr1, Kcnn4, Ncam1, Robo1*
cell adhesion (0)
  *Cdh2, Lama1, Nrxn1, Pcdhb1*
regulation of localization (0)
  *Axin2, Fzd1, Jak2, Rab3a, Wnt5a*

**PRC Acquired**

regulation of cell proliferation (0)
  *Nodal, Notch1, Gli1, Sox2, Erbb3*
biological adhesion (0)
  *Axl, Cd47, Fzd7, Lama4, Src*
regulation of cell shape (0.0015)
  *Icam1, Anxa7, Coro1a, Ezr, Myh10*
anterior/posterior axis specification (0.0025)
  *Fzd5, Frs2, Tcf7l1, Tdgf1*

**Figure 3.  PRC occupancy at gene promoters is highly dynamic throughout all stages of the neuronal differentiation.**

A   Numbers of PRC$^+$ promoters from ESC to day 30. In terminally differentiated neurons, 2,178 genes are marked by H3K27me3.
B   Dynamic changes in H3K27me3 presence at gene promoters. Vertical lines represent each of the promoters marked by H3K27me3$^+$ in at least one time point. Throughout differentiation, 1,408 genes are marked by H3K27me3 (PRC Maintained), 2,699 lose H3K27me3 (PRC Lost), and 1,565 gain H3K27me3 (PRC Acquired). Color represents number of reads in transcription start site (TSS) window, scaled for visualization.
C   Examples of enriched GO terms and gene examples among the groups of genes classified as PRC Maintained, Lost, and Acquired, using as background all genes.

Schimmelmann *et al*, 2016), functional endocrine cells (1,273 genes; Xie *et al*, 2013), and in E12.5 heart ventricle apex (1,052 genes; He *et al*, 2012).

To study the dynamics of Polycomb repression across the different stages of differentiation, we grouped promoters according to H3K27me3 presence or absence (Fig 3B). H3K27me3 marks a total of 5,672 genes in at least one time point (Dataset EV2). Over half of the H3K27me3$^+$ genes in ESCs lose H3K27me3 between days 1 and 30 (*"PRC lost" genes*), while 1,408 genes remain H3K27me3$^+$ throughout the differentiation timeline (*"PRC maintained"*; Fig 3C). Genes that lose H3K27me3 include *Fos, Zic1, Ncam1,* and *Wnt5a*, and are enriched in GO terms such as *"nervous system development"*, *"regulation of signaling"*, and *"cell adhesion"*, consistent with their activation upon acquisition of the neuronal phenotype. Genes associated with PRC in all time points include developmental transcription factors such as *Nkx2-5*, *Pax6,* and *Gata4*, and are enriched in functions such as *"anatomical structural development"*, *"pattern specification process"*, *"regulation of cell differentiation"*, and *"DNA dependent transcription"*. *De novo* acquisition of H3K27me3 is detected throughout differentiation (*"PRC acquired"* genes), with a transient appearance of H3K27me3 at 931 genes between days 1 and 3, and at 351 genes in post-mitotic neurons (Fig 3C). These genes are enriched in GO terms such as *"regulation of cell proliferation"* and include important signaling regulators as *Nodal, Notch1,* and *Fzd5*. Our results show that genes targeted by PRC during neuronal differentiation undergo dynamic changes at appropriate stages of differentiation.

## Poised RNAPII-S5p is present at H3K27me3$^+$ promoters throughout differentiation

To investigate whether poised RNAPII-S5p is lost from Polycomb-repressed promoters upon exit from pluripotency, or associated with Polycomb throughout terminal differentiation, we produced ChIP-seq datasets for RNAPII-S5p and RNAPII-S7p in ESCs and at days 1, 3, 16, and 30 (Fig 4A). Inspection of single-gene profiles revealed different promoter states and dynamic changes (Fig 4B). For example, the housekeeping gene *Eif2b2* is always expressed at the mRNA level and occupied by RNAPII-S5p and RNAPII-S7p, whereas testis-specific *Brs3* is always inactive and lacks H3K27me3 and RNAPII-S5p and RNAPII-S7p throughout our differentiation timeline. In contrast, *Adcy5*, a gene implicated in axonal elongation and in addiction (del Puerto *et al*, 2012; Procopio *et al*, 2013), is marked by RNAPII-S5p and H3K27me3 in ESCs and becomes Active from day 16, acquiring S7p, losing H3K27me3, and maintaining S5p. *Pkp1*, which has roles in epidermal morphogenesis (South *et al*,

2003), is always marked by H3K27me3, but associated with poised RNAPII-S5p only in ESCs and in early differentiation. Notably, *Pitx1*, a transcription factor with roles in limb morphogenesis (Infante *et al*, 2013), is marked by RNAPII-S5p and H3K27me3 at all five time points, without S7p or mRNA expression. Examples of *de novo* PRC repression include *Ajuba,* a gene implicated in cell-cycle progression (Hirota *et al*, 2003) which is downregulated during differentiation while acquiring H3K27me3, and retaining S5p, S7p, and low levels of mRNA expression. Genes marked with H3K27me3, RNAPII-S5p, and RNAPII-S7p in bulk ChIP assays have been previously described in ESCs and shown to correspond to promoter fluctuations between active and PRC-repressed states in different cells or alleles (Brookes *et al*, 2012; Kar *et al*, 2017).

For an unbiased exploration of the transitions between states of Polycomb repression and productive transcription across differentiation, we identified the promoters that are associated with RNAPII or Polycomb in each time point. RNAPII-S5p marks both active and PRC-repressed promoters, RNAPII-S7p marks only at active promoters, and H3K27me3 marks PRC-repressed promoters. As previously reported, most promoters marked by H3K27me3 in ESCs are also bound by RNAPII-S5p (81%; Fig 4C; see also Brookes *et al*, 2012). Importantly, we find that many promoters are positively marked by RNAPII-S5p and H3K27me3 throughout differentiation, showing that RNAPII poising at PRC-marked promoters is a general property of Polycomb repression and not only specific to ESCs. During terminal differentiation, PRC/S5p genes become increasingly marked by RNAPII-S7p (Fig 4D) and expressed at the mRNA level (Fig 4E), suggesting a role for PRC repression not limited to strict gene silencing, but also in modulating gene expression levels, as seen previously in ESCs (Brookes *et al*, 2012; Kar *et al*, 2017).

## Genome-wide transitions in Polycomb repression states

To understand the dynamic transitions in promoter states throughout differentiation, we classified genes according to the presence of RNAPII-S5p, RNAPII-S7p, and H3K27me3, in each time point (Fig 5A). PRC-positive promoters were classified as "PRC Only" (H3K27me3$^+$S5p$^-$S7p$^-$), as "PRC/S5p" (H3K27me3$^+$S5p$^+$S7p$^-$), and as "PRC/Active" (H3K27me3$^+$S5p$^+$S7p$^+$). PRC-negative promoters were classified as "Active" (H3K27me3$^-$S5p$^+$S7p$^+$), or as "Inactive" (H3K27me3$^-$S5p$^-$S7p$^-$). The other three combinations of promoter marks (S7p Only, S5p Only, and PRC/S7p) were uncommon and not explored further in the present study. The numbers of genes in each state are summarized in Fig EV4A; gene promoter states are listed in Dataset EV2. We found that the decrease in the number of promoters marked by PRC is accompanied by an increase in Active

**Figure 4. RNAPII-S5p overlaps with Polycomb at all stages of differentiation.**

A   Schematic summarizing the strategy used to define promoter states through differentiation based on the interplay between PRC and RNAPII.

B   Examples of single-gene ChIP-seq profiles across differentiation. *Eif2b2* and *Brs3* are always Active and Inactive, respectively. *Adcy5* and *Pkp1* are PRC/S5p in ESC and days 1/3, but become Active or PRC Only, respectively. *Pitx1* maintains the poised PRC/S5p conformation from ESCs until day 30 neurons. *Ajuba* is Active in ESCs but acquires H3K27me3 at day 16 while becoming downregulated.

C   Proportion of H3K27me3$^+$ promoters (dark red) marked by RNAPII-S5p (purple) throughout differentiation. The vast majority of H3K27me3$^+$ promoters are bound by RNAPII-S5p in all time points.

D   Proportion of H3K27me3$^+$ and S5p$^+$ promoters (purple) marked by S7p (green) throughout differentiation. The number of promoters marked by H3K27me3$^+$S5p$^+$S7p$^+$ increases during differentiation.

E   Proportion of H3K27me3 and RNAPII-S5p promoters (purple) that are also transcribed (gold; TPM > 1). The number of actively transcribed promoters increases during differentiation, in agreement with the increased presence of S7p.

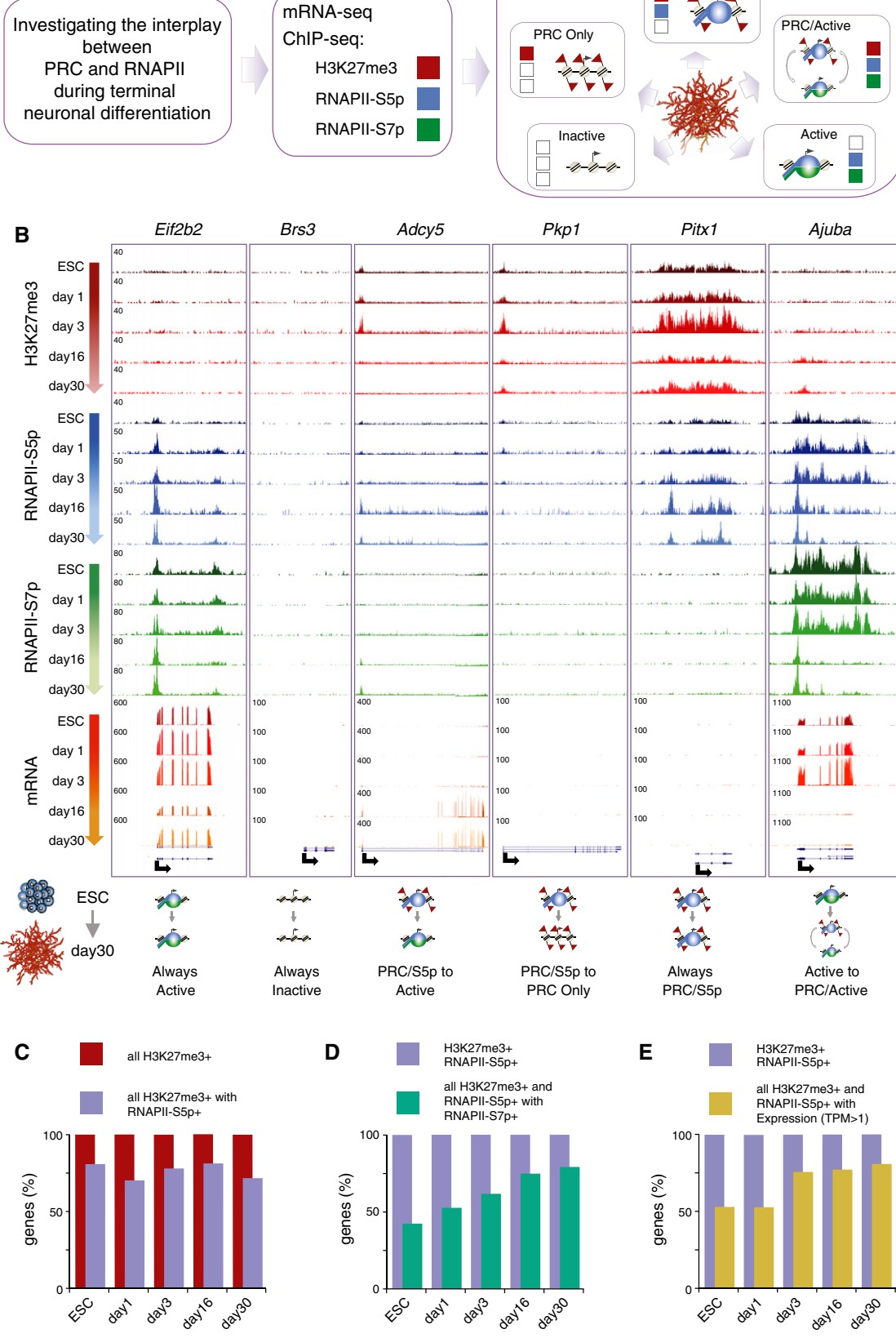

**Figure 4.**

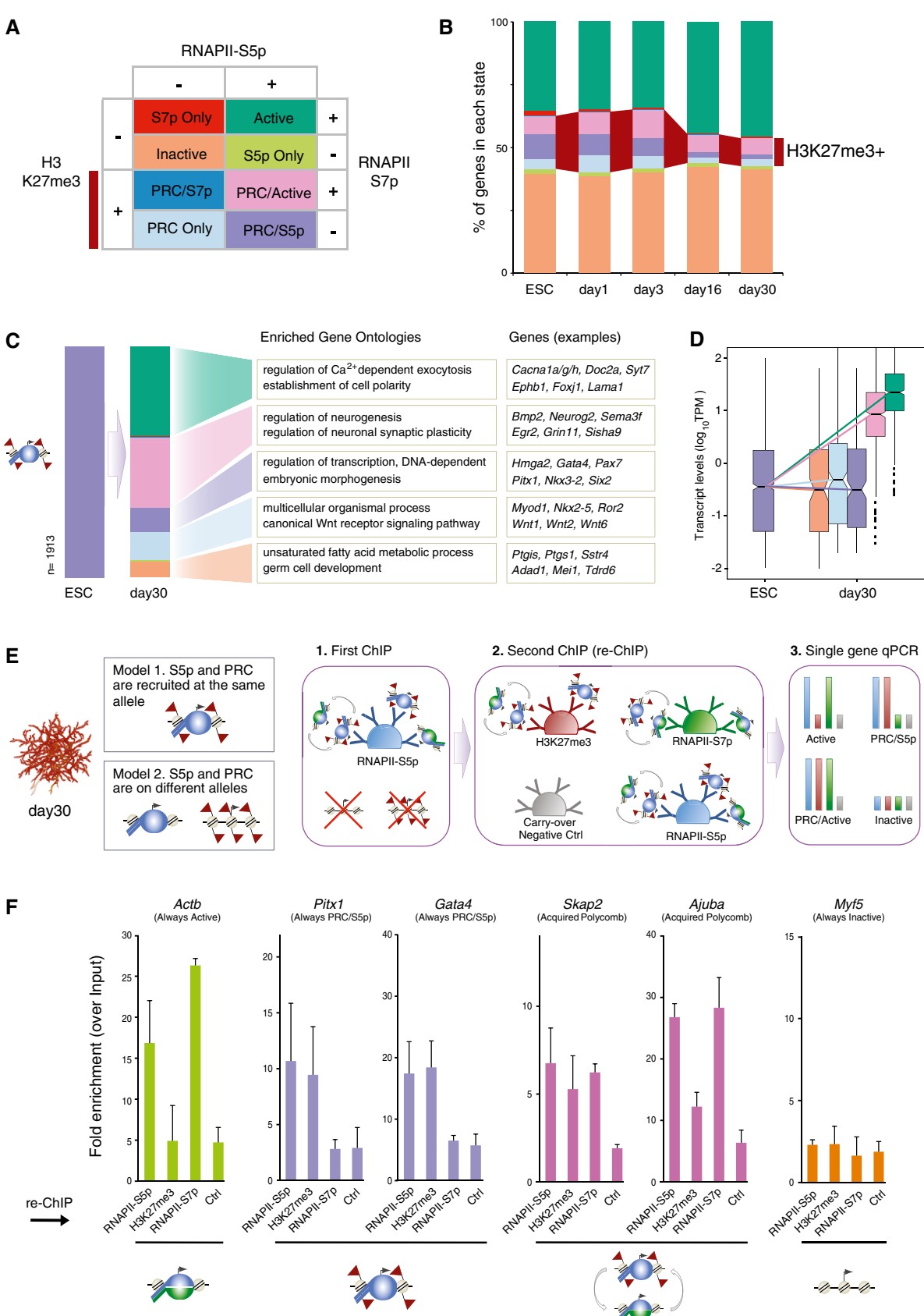

**Figure 5.**

promoters, while the number of Inactive promoters remains stable (Fig 5B). The decrease in PRC-positive promoters occurs across all PRC classes, except for a relative increase in the PRC/Active state (Fig 5B). By following the changes in promoter state between time points (Fig EV4B), we show that the increase in the Active promoter state in days 16 and 30 originates mostly from genes that were PRC repressed in ESCs to day 3; this transition is accompanied by increased mRNA expression, as expected from the acquisition of S7p and loss of H3K27me3 at this group of genes (Fig EV4C). These genes with newly Active promoter states have specific neuronal functions (Fig EV4D).

Next, we investigated the promoter state transitions of the genes associated with the poised PRC/S5p promoter state in ESCs (Fig 5C). Many of the PRC/S5p genes in ESCs become Active (743/1913) or PRC/Active (580/1913) in day 30, and have functions in cell polarity or synaptic activity. Acquisition of S7p is accompanied by increased mRNA expression, as expected (Fig 5D). Notably, a group of PRC/S5p genes in ESCs retain the poised conformation in day 30 neurons (200/1,913 genes; Fig 5C). These genes are important developmental transcription factors (TFs) which regulate non-neuronal cell lineages (e.g., *Gata4*, *Pax7*, *Nkx3-2*). We also find PRC/S5p genes that resolve to Inactive (129/1,913), with functions in "*germ cell development*", or that become PRC Only (236/1,913), with functions including "*Wnt signalling*".

Other interesting Polycomb-dependent transitions in promoter activation states are those that result in *de novo* acquisition of H3K27me3 in either days 1/3 or days 16/30 (Fig EV5A and B). Most genes that gain H3K27me3 in days 1 or 3 tend to be Active in ESCs, and go back to their Active promoter state in day 30, with the acquisition of H3K27me3 in days 1/3 being accompanied by decreased gene expression (Fig EV5A). The genes that acquire PRC at days 16 or 30 are downregulated at this stage of differentiation, and tend to be Active in ESCs (Fig EV5B).

Taken together, our results show a role for Polycomb, most often accompanied by poised RNAPII-S5p complexes, in the silencing of non-neuronal TFs in neurons and in fine-tuning gene expression states during differentiation.

### H3K27me3 chromatin is co-occupied with RNAPII-S5p complexes in mature neurons

To test the simultaneous occupancy of RNAPII-S5p and H3K27me3 at Polycomb-repressed genes in terminally differentiated neurons, we performed sequential ChIP (or re-ChIP; Fig 5E). We immunoprecipitated chromatin from day 30 neurons with the antibody used to map RNAPII-S5p, before re-immunoprecipitation with different antibodies. We show that the promoter of the Active gene *Actb* is co-bound by RNAPII-S5p and RNAPII-S7p, but not by H3K27me3, as expected (Fig 5F). Notably, the promoters of non-neuronal transcription factors *Pitx1* and *Gata4*, which are classified as PRC/S5p in all time points, are co-occupied in neurons by H3K27me3 and RNAPII-S5p, but not RNAPII-S7p. These results show that RNAPII-S5p and H3K27me3 simultaneously occupy Polycomb-repressed promoters in mature dopaminergic neurons. We also tested the co-association of H3K27me3 with RNAPII-S5p at *Skap2* and *Ajuba,* which are PRC/Active genes that acquire H3K27me3 *de novo* in day 30, and found that H3K27me3 and RNAPII-S5p also co-localize at these promoters. In agreement with the PRC/Active classification, RNAPII-S5p co-immunoprecipitates with RNAPII-S7p, consistent with the presence of the active (S5p$^{+}$S7p$^{+}$) form of RNAPII at some alleles or in different cells (Brookes *et al*, 2012; Kar *et al*, 2017). None of the marks tested were enriched at the inactive gene (*Myf5*), confirming the specificity of the assay. Re-ChIP with an unrelated antibody, against plant steroid digoxigenin, was used as a negative control to confirm minimal antibody carryover from the first to the second immunoprecipitation. The sequential ChIP analyses performed here in terminally differentiated neurons show that RNAPII poising is present at PRC-repressed genes in post-mitotic neurons.

### Poised PRC/S5p promoter states are associated with increased potential for derepression upon Polycomb knockout or chemical inhibition

To explore the biological relevance of the poised PRC/S5p promoter state in ESCs, neuronal progenitors, and neurons, we investigated the effect of Polycomb knockout at PRC/S5p genes compared to PRC Only genes. First, we took advantage of published transcriptome resources in ESCs which measured transcriptional derepression after stable knockout of PRC2 component *Eed* or conditional knockout of PRC1 component *Ring1B* in *Ring1A*$^{-/-}$ background (Endoh *et al*, 2008; Cruz-Molina *et al*, 2017). We find that a larger proportion of PRC/S5p genes are upregulated in both PRC knockout cell lines, compared to PRC Only genes (Fig 6A). Only a small fraction of Active and Inactive genes responded to PRC knockout, which may arise through more indirect regulatory effects associated with adaptation in the knockout cell lines. Second, we performed similar analyses using a published microarray dataset from an *in vivo* PRC2

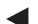

**Figure 5.  RNAPII-S5p co-localizes with H3K27me3 in terminally differentiated neurons.**

A    Schematic of gene classification according to the occupancy of H3K27me3, S5p, and S7p at promoter regions (2-kb windows centered on Transcription Start Sites; TSSs).

B    Proportion of genes with each promoter state during terminal neuronal differentiation. Dark red: H3K27me3$^{+}$ genes.

C    PRC/S5p promoters in ESCs and their promoter state in day 30 neurons. Examples of enriched GO terms (determined using as background all PRC/S5p genes in ESCs) and genes in each category.

D    Expression level of PRC/S5p promoters in ESCs and day 30 neurons. Acquisition of S7p correlates with increased expression level. Genes with PRC/S5p, PRC Only, or Inactive state remain repressed in day 30 neurons.

E    Sequential ChIP (re-ChIP) was performed using day 30 neurons to test RNAPII and H3K27me3 co-occupancy, as described in the schematic. The first ChIP with anti-RNAPII-S5p antibody was followed by a second ChIP using antibodies against RNAPII-S5p (positive control), H3K27me3, RNAPII-S7p, or Digoxigenin (carryover negative control "Ctrl").

F    Sequential ChIP shows RNAPII-S5p co-association with H3K27me3 at PRC/S5p (*Pitx1* and *Gata4*) and PRC/Active (*Skap2* and *Ajuba*) promoters but not at Active (*Actb*) or Inactive (*Myf5*) promoters. The mean and SD are from two biological replicates.

knockout in GABAergic striatal medium spiny neurons which results in neurodegeneration (von Schimmelmann *et al*, 2016). Although the neuronal subtypes are not fully matched between the promoter state classifications from our neuronal cultures, that simulate the ventral tegmental area, and the GABAergic neurons from the striatum, we find that most genes upregulated upon PRC2 loss in *ex vivo* striatal neurons have the PRC/S5p promoter state in our mature day 30 neurons (52 out of the total 150 upregulated genes; hypergeometric test for enrichment, $P = 3.1 \times 10^{-56}$; of note, 39 of those are PRC/S5p at all stages; Fig 6B). In contrast, only 17 genes upregulated in the striatal neurons are classified as PRC Only in the dopaminergic cultures ($P = 6.8 \times 10^{-7}$), and a smaller proportion of Active or Inactive genes become upregulated. These results suggest that the PRC/S5p state leads to increased likelihood for derepression upon Polycomb loss in both ESCs and differentiated neurons, which has implications for our understanding of Polycomb roles in maintaining cell differentiation states.

Lastly, we tested the response of PRC/S5p and PRC Only genes to PRC2 enzymatic inhibition at three different stages of differentiation (ESC, day 3, and day 30). We used UNC1999, a chemical drug that blocks the enzymatic activity of both EZH1 and EZH2 (Konze *et al*, 2013), in two concentrations (1 and 3 μM), and measured relative RNA levels by quantitative PCR. We tested six genes that maintain the PRC/S5p state throughout differentiation (*Tal1*, *Gata4*, *Skor1*, *Pitx1*, *HoxA7*, and *Mesp1*), three genes that are PRC/S5p in ESCs but become PRC Only from day 3 onwards (*Pouf3*, *Pkp1*, *She*) and three genes that maintain the PRC Only promoter state throughout differentiation (*Fam150b*, *Fgf20*, *Chdrl2*). We find that five out of the six PRC/S5p genes that maintain their promoter state are upregulated in ESCs, day 3 and day 30 upon enzymatic inhibition of PRC2, with increased derepression at the highest inhibitor concentration (Fig 6C). In contrast, none of the analyzed genes acquiring or maintaining the PRC Only state show detectable RNA levels or respond to EZH1/2 inhibition at either UNC1999 concentration in any of the three time points tested. Quality of PCR primers was verified by agarose gel electrophoresis using genomic DNA (Appendix Fig S1).

Taken together, these results suggest that the poised RNAPII-S5p is associated with increased potential for derepression upon loss of Polycomb repression in ESCs and in differentiated cells at all stages of maturation, including terminally differentiated neurons both *in vitro* and *in vivo*.

### Major non-neuronal transcription factors maintain PRC/S5p occupancy upon terminal differentiation

To investigate further the significance of the poised PRC/S5p promoter state during neuronal cell commitment, we focused on 115 genes that maintain the PRC/S5p state during all the sampled stages of neuronal differentiation (Fig 7A). For comparison, we chose two other groups of genes which are also PRC/S5p in ESCs but remain silent in neurons through other promoter states; one group becomes PRC Only (losing S5p; 127 genes), and the other becomes Inactive (losing both H3K27me3 and S5p; 92 genes; Figs 7A and EV6A). Average profiles of H3K27me3 and RNAPII-S5p occupancy show that these three groups of genes are clearly marked by both H3K27me3 and RNAPII-S5p in ESCs (Fig 7B; Brookes *et al*, 2012). The genes that maintain the PRC/S5p promoter state throughout neuronal differentiation have similar S5p occupancy at their promoters and throughout the coding region in ESCs, but show higher average levels of H3K27me3 occupancy in comparison with genes that resolve to PRC Only or Inactive (Fig 7B). We noticed that the PRC/S5p genes that maintain their state during neuronal differentiation are associated with wider regions of H3K27me3 occupancy compared to genes that lose RNAPII-S5p or to all other neuronal H3K27me3$^+$ promoters in both ESCs and day 30 neurons (Fig 7B–D).

To explore the biological relevance of the three groups of genes, we performed GO enrichment analyses (Fig EV6B). Interestingly, the group of 115 PRC/S5p genes that maintain their state is enriched in TFs important for non-neuronal lineages such as *Gata4*, *Mesp1*, *Tbx2* (GO term: "*heart morphogenesis*"), *Barx1*, *Gata6*, *Hoxb5* ("*epithelial cell differentiation*"), and *Osr2*, *Pax1*, *Tcfap2a* ("*bone morphogenesis*"), suggesting a role for the poised PRC/S5p state in terminally differentiated neurons in maintaining cell identity and plasticity by repressing non-neuronal genes.

### Key non-neuronal transcription factors that maintain the PRC/S5p state in neurons have high CpG content

Polycomb occupancy in ESCs is highly correlated to CpG content (Ku *et al*, 2008; Deaton & Bird, 2011) and unmethylated CpGs (Sen *et al*, 2016). To test whether the retention of the PRC/S5p state throughout neuronal differentiation is related to high CpG content, we measured the proportion of the promoter regions covered by CpG islands, the GC content of TSSs, and the extent of CpG island coverage of gene bodies. We observe that the promoter of PRC/S5p genes that maintain the poised state in neurons is more extensively covered by CpG islands than the genes that become PRC Only or Inactive (Fig 7E). Their promoter GC content is also higher (Fig EV6C), and the CpG islands cover longer portions of gene body (Fig EV6D) than at genes that become Inactive. These results suggest that Polycomb repression in co-association with poised RNAPII-S5p correlates with specific sequence features of gene promoters and gene bodies, namely with high CpG content.

---

**Figure 6.   Poised PRC/S5p promoter states have increased potential for reactivation upon loss of Polycomb repression.**
   A   Effect of PRC deletion in ESCs (PRC2 component EED and PRC1 components RING1B and RING1A). Barplots represent the percentage of genes which are upregulated more than twofold according to their promoter state in ESCs.
   B   Effect of PRC2 deletion in striatal GABAergic medium spiny neurons in 6-month-old mice. Barplot represents the percentage of genes which are upregulated more than twofold according to their promoter state in day 30 neuron.
   C   RNA levels of PRC/S5p genes were measured by qRT–PCR upon treatment with EZH1/2 inhibitor UNC1999 at 1 and 3 μM, for 24 h prior to RNA extraction. Relative levels are normalized to *Actb* internal control. Three or two biological replicates were performed in ESC and day 3 or in day 30 neurons, respectively. Black lines connect mean values.

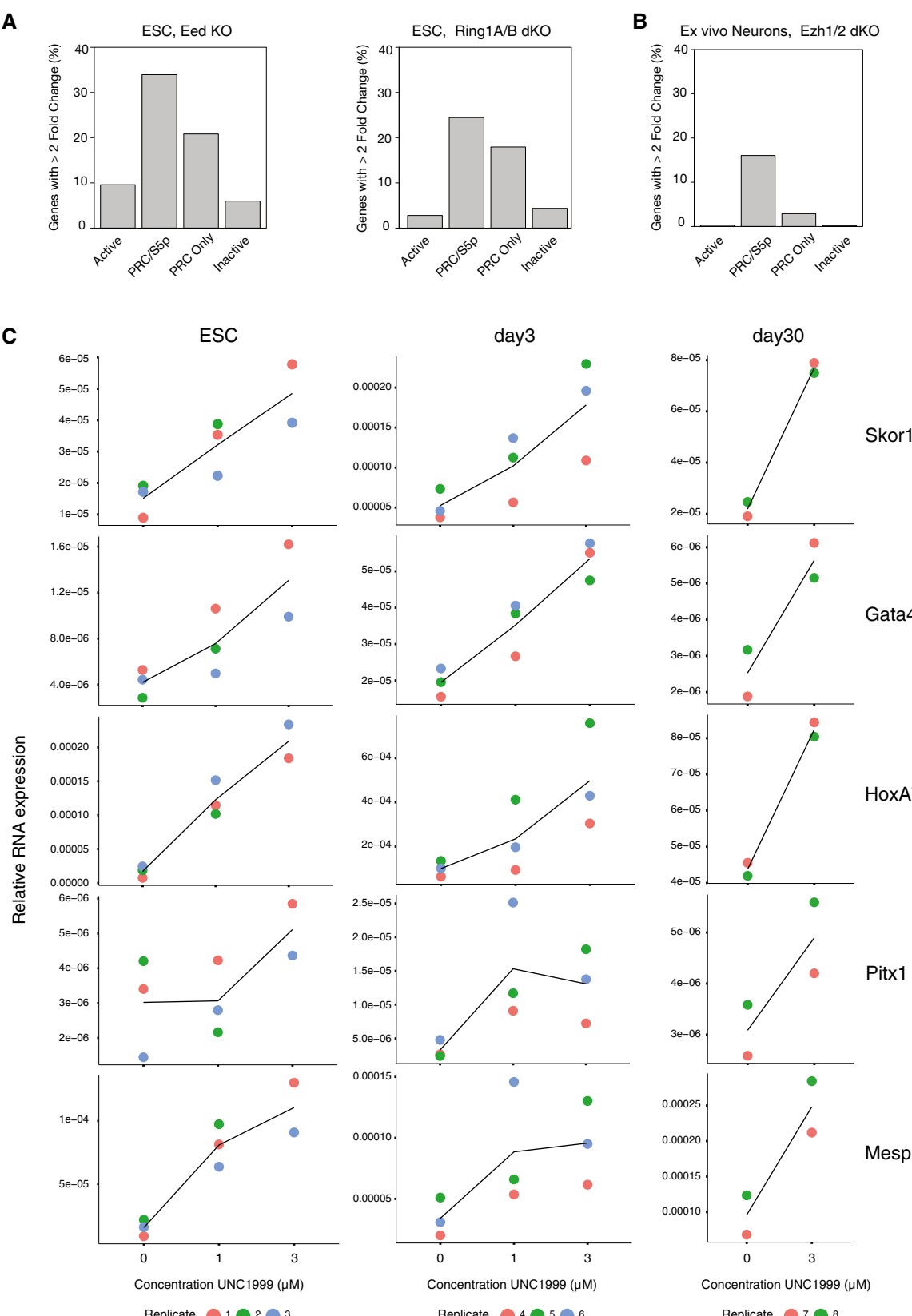

**Figure 6.**

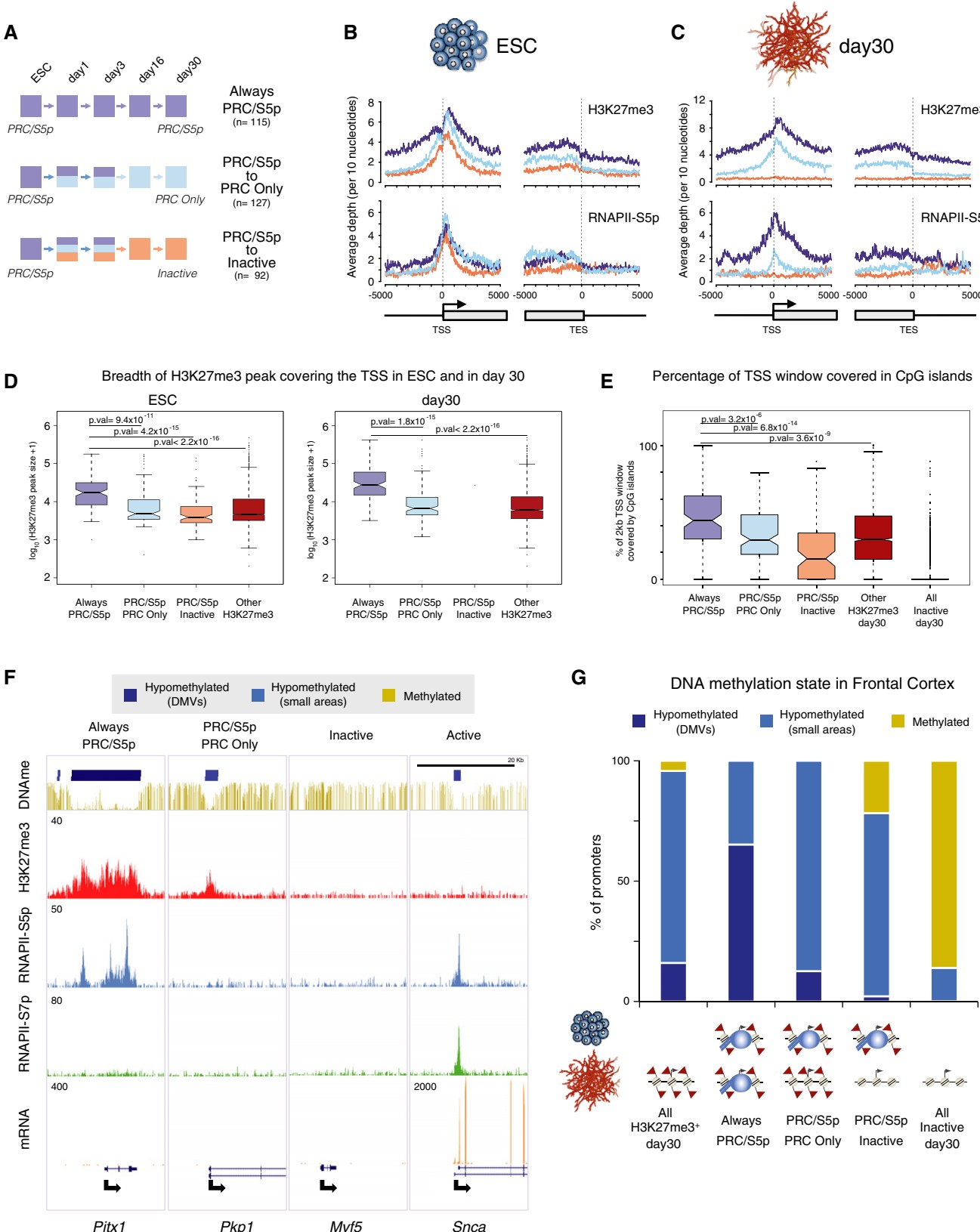

**Figure 7.**

**Figure 7.   PRC/S5p genes that maintain the poised state throughout differentiation exhibit broader distributions of H3K27me3 and RNAPII-S5p which coincide with DMVs.**

A   PRC/S5p genes are kept silenced through different mechanisms during neuronal differentiation by maintaining PRC/S5p (Always PRC/S5p; purple), resolving to PRC Only (light blue), or resolving to Inactive (orange). Colors indicate promoter states in each time point.

B   Average ChIP-seq enrichment plots of H3K27me3 and S5p in 10-kb genomic windows centered on transcription start site (TSS) and transcription end sites (TES), in ESCs.

C   Average ChIP-seq enrichment plots of H3K27me3 and S5p in 10-kb genomic windows centered on TSS and TES, in day 30 neurons.

D   Boxplot of the length of the genomic regions enriched for H3K27me3 that overlap with promoters in ESC or in day 30 neurons. The length of H3K27me3-enriched regions covering the remaining H3K27me3$^+$ genes is shown for comparison. *P*-values are calculated with Wilcoxon rank-sum test.

E   Percentage of TSS window covered by CpG islands of PRC/S5p that maintain their state is significantly larger compared to the other gene subgroups. *P*-values are calculated with Wilcoxon rank-sum test.

F   UCSC Browser tracks of genes classified in day 30 neurons as Always PRC/S5p, PRC Only, Inactive, and Active genes. First row shows methylated (gold) or hypomethylated DNA methylation valleys (DMVs, dark blue; and smaller regions of hypomethylated DNA, light blue) in mouse frontal cortex.

G   Proportion of promoters in the three gene groups which overlap with methylated and hypomethylated DNA. Always PRC/S5p genes are preferentially located at DMVs. All H3K27me3$^+$ and Inactive genes in day 30 neurons are shown for comparison.

## Key non-neuronal transcription factors that maintain the PRC/ S5p state in neurons are located in DNA methylation valleys

Previous reports have identified DMVs (wide depletions of DNA methylation, longer than 5 kb) in different cell lineages, which coincide with CpG-rich regions often located at loci encoding for developmental transcription factors (Xie *et al*, 2013). To explore the association of genes that maintain the poised PRC/S5p conformation throughout differentiation with DMVs, we mined a published dataset for DNA methylation from mouse frontal cortex (Lister *et al*, 2013), and classified DMVs (Xie *et al*, 2013). Visual inspection of single-gene profiles shows that the regions occupied by H3K27me3 and RNAPII-S5p in neurons often coincide with large stretches of hypomethylated DNA, as shown for *Pitx1* (Fig 7F). Expanding these observations genome-wide, we find that all genes with PRC/S5p promoters that maintain their poised state throughout neuronal differentiation are hypomethylated in mouse cortex, with the majority (65%; 70/115) overlapping with DMVs (Fig 7G). In contrast, only 13% (16/127) of genes that become PRC Only or 2% (2/92) of genes that become Inactive reside in DMVs. Therefore, the maintenance of the PRC/S5p state is a property of important developmental transcription factors that are also rich in CpG and lack DNA methylation in terminally differentiated tissues.

## Maintenance of PRC/S5p state occurs at the promoters of developmental transcription factors that are candidate drivers of trans-differentiation

The observation that key developmental TFs are repressed in postmitotic neurons through Polycomb repression in the presence of poised RNAPII-S5p led us to hypothesize that this conformation contribute to a state that makes neurons permissive to losing their identity and potentially to trans-differentiation to other cell types. To explore this hypothesis, we took advantage of Mogrify, a predictive system of the TFs necessary to induce cell-type reprogramming, which combines gene expression data with regulatory network information (Rackham *et al*, 2016; Fig 8A). We overlapped the TFs that Mogrify predicts to be important for trans-differentiation to various cell fates with the promoter states that we have identified in day 30 neurons. We find that the Mogrify lists of candidate genes for trans-differentiation toward non-neuronal lineages, such as cardiomyocytes, skin fibroblasts, and skeletal muscle, are significantly enriched in genes marked by H3K27me3 in day 30 neurons, and especially in genes with the poised PRC/S5p promoter state (Fig 8B; for additional examples see Fig EV7A). In contrast, the Mogrify lists of reprogramming TFs are depleted of genes that have Inactive promoter states in day 30 neurons. We also investigated the lists of reprogramming TFs predicted to be important for trans-differentiation to neuronal lineages, including to hippocampus, neuron, or brain. These genes are more often Active or PRC/Active in day 30 neurons, unsurprisingly due to their expression in the neuronal lineage, or alternatively associated with poised PRC/S5p promoter state (Fig 8B).

Mogrify also ranks TFs according to their importance for trans-differentiation when they are predicted to regulate a larger number of the genes required to drive changes in phenotype; these genes are scored as having *high influence* on trans-differentiation. In contrast, the genes that are found to be less important for trans-differentiation (and scored as *low influence*) tend to be less connected or to regulate genes that are less specific to the target cell type (see schematic

**Figure 8.   Genes with the poised PRC/S5p promoter state in terminally differentiated neurons are enriched for TFs with trans-differentiation potential.**

A   Cells can trans-differentiate into other cell types by forced expression of lineage-specific TFs. The Mogrify algorithm identifies candidate TFs that drive cell conversion as exemplified in the schematic (Rackham *et al*, 2016).

B   Promoter state of TFs in day 30 which are predicted by Mogrify to drive trans-differentiation toward non-neuronal or neuronal cell types. H3K27me3$^+$ promoters are enriched in the Mogrify lists of trans-differentiation candidates. PRC/S5p and the PRC/Active genes are the mostly enriched promoter classes. Inactive genes are depleted. Enrichment and depletion were tested with the hypergeometric test.

C   Mogrify ranks all TFs by the likelihood to induce conversion from cell types A to B. By averaging these ranks, it is possible to identify those TFs most frequently required for a conversion into a certain cell type. The ranks are used to estimate the importance of a given TF to drive the conversion.

D   TFs important for driving trans-differentiation toward different cell types were grouped according to their promoter state and tested for enrichment in high or low importance factors (upwards/downwards arrows). Purple and white backgrounds correspond to enrichment tested for promoter states in day 30 or maintained along all differentiation, respectively. Enrichment in high or low importance was tested with Gene Set Enrichment Analysis, considered significant when false discovery rate (FDR) was < 25%.

                                        

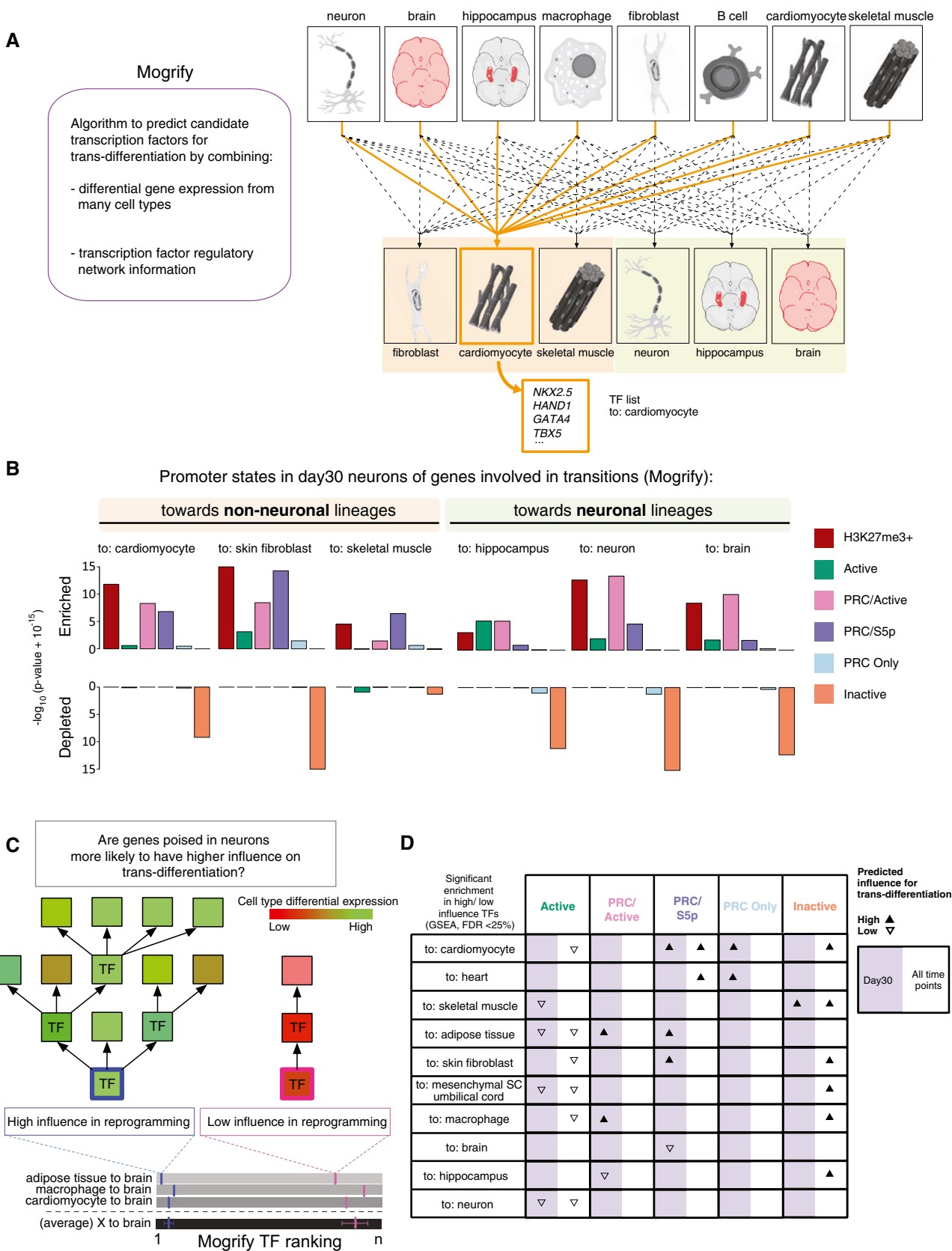

**Figure 8.**

in Fig 8C). To test whether genes with *high influence* are enriched in PRC-repressed promoter states, we used Gene Set Enrichment Analysis (GSEA; Subramanian *et al*, 2005). Interestingly, we find that genes that have Active promoter states in day 30 neurons are significantly enriched for *low-influence* TFs (false discovery rate < 25%), whereas poised PRC/S5p genes are significantly enriched in *high-influence* TFs which are candidates for promoting conversions toward non-neuronal cell fates (Fig 8D; see example in Fig EV7B). These results show that H3K27me3 and RNAPII-S5p mark silent genes in neurons which are predicted to be important to promote trans-differentiation to non-neuronal lineages, suggesting a role of the poised RNAPII-S5p state in allowing cell identity conversions and of Polycomb repression in stabilizing specific differentiated states.

## Discussion

Most genes repressed by Polycomb in ESCs are kept in a poised state through their association with an unusual form of RNAPII characterized by the S5p modification (Stock *et al*, 2007; Brookes *et al*, 2012; Tee *et al*, 2014; Ma *et al*, 2016). The PRC/S5p promoter state is thought to play an important role in the regulation of cell plasticity, conferring pluripotent stem cells with the ability to rapidly activate specific subgroups of genes in response to developmental stimuli (Brookes & Pombo, 2009; Voigt *et al*, 2012; Di Croce & Helin, 2013). However, how the interplay between Polycomb and poised RNAPII evolves upon differentiation has remained unexplored. Here, we optimized an *in vitro* differentiation approach to obtain functional dopaminergic neurons at different stages of maturation, and produced a comprehensive collection of datasets that provides an in depth profile of the transcriptional and promoter state dynamics during differentiation. We observe that promoter state transitions during differentiation are instructed by a tight interplay between RNAPII poising and Polycomb repression. We show for the first time that poising of genes by RNAPII-S5p occurs at intermediate and terminal stages of differentiation, that it associates with increased potential for derepression upon loss of Polycomb repression, and that it may have roles in trans-differentiation events *in vivo* or *in vitro*.

We show *de novo* acquisition of PRC marks at many promoters during differentiation, which coincides with downregulation of gene expression. Notably, we identify a subgroup of genes that maintain the poised PRC/S5p conformation throughout all stages of differentiation studied, including many transcription factors important for specification of non-neuronal lineages. We directly demonstrate the simultaneous co-association of H3K27me3 and RNAPII-S5p at PRC/S5p and PRC/Active genes in differentiated neurons using sequential ChIP, both at genes that maintain the PRC/S5p state and that acquire PRC *de novo*.

We observe that the genes that maintain the PRC/S5p state throughout differentiation have a specific chromatin structure characterized by high CpG content, broad distributions of RNAPII-S5p and H3K27me3, and overlap with large hypomethylated regions or DMVs. This cohort of genes is enriched for developmental factors with key roles in non-neuronal lineages, such as heart, bone, or epithelium. Differentiated cells can be reprogrammed and their identities altered in a process called trans-differentiation (Ladewig *et al*, 2013). To explore the implication of maintaining genes that encode

for lineage-commitment TFs in a poised Polycomb-repressed configuration throughout differentiation, we considered whether they are candidate mediators of trans-differentiation. We discovered that the TFs that maintain the PRC/S5p promoter state during differentiation are important candidate drivers of trans-differentiation conversions. Future studies in other terminally differentiated cell types will help understand whether the poised chromatin state is an inherent state of repression for these important transcription factors, or alternatively whether distinct sets of poised PRC/S5p genes are specifically chosen in each specific cell type to restrict the potential to selectively reprogramming into another cell type, either during the normal physiology of a tissue or in disease.

In conclusion, our study identifies key roles of Polycomb repression and RNAPII-S5p poising in modulating the dynamic changes in gene expression that occur during ESC terminal differentiation into dopaminergic neurons, and provides comprehensive maps of gene expression and promoter states of repression which are a valuable resource for regenerative and developmental biology. Studying gene regulation in terminally differentiated neurons with a dopaminergic phenotype is of particular interest for understanding the biology behind Parkinson's disease and for the generation of cells for replacement therapy as possible treatment.

## Materials and Methods

### Cell culture and differentiation

Cells were grown at 37°C in a 5% (v/v) $CO_2$ incubator. Mouse embryonic stem cells (ESC-46C; ES cell line E14tg2a that expresses GFP under Sox1 control; Ying *et al*, 2003), kindly provided by Domingos Henrique, were grown in GMEM medium (Invitrogen, Cat# 21710025), supplemented with 10% (v/v) Fetal Calf Serum (FCS; BioScience LifeSciences, Cat# 7.01 batch number 110006), 2,000 U/ml LIF (Millipore, Cat# ESG1107), 0.1 mM β-mercaptoethanol (Invitrogen, Cat# 31350-010), 2 mM L-glutamine (Invitrogen, Cat# 25030-024), 1 mM sodium pyruvate (Invitrogen, Cat# 11360039), 1% penicillin-streptomycin (Invitrogen, Cat# 15140122), 1% MEM Non-Essential Amino Acids (Invitrogen, Cat# 11140035) on gelatin-coated (0.1% v/v) Nunc T25 flasks. The medium was changed every day, and cells were split every other day. Before sample collection for ChIP-seq and mRNA-seq, ESCs were plated on gelatin-coated (0.1% v/v) Nunc 10-cm dishes in serum-free ESGRO Complete Clonal Grade Medium (Millipore, Cat# SF001-B) to which was added 1,000 U/ml LIF. Cells were grown for 48 h, with a medium change at 24 h. ESC batches were tested for mycoplasma infection.

Early neuronal differentiation was optimized based on the method described in Abranches *et al* (2009). ESCs were plated with high density ($1.5 \times 10^5$ cells/cm$^2$) in serum-free ESGRO Complete Clonal Grade Medium (Millipore, Cat# SF001-B) to which 1,000 U/ml LIF was added. After 24 h, ESCs were washed 3 times with PBS without magnesium and calcium, incubated in PBS for 3 min at room temperature, and then dissociated by incubating in 0.05% (v/v) Trypsin (Gibco, Cat# 25300-054) for 2 min at 37°C. ESCs were plated onto 0.1% (v/v) gelatin-coated 10-cm dishes (Nunc) at $1.6 \times 10^6$ cells/dish in RHB-A (Takara-Clontech, Cat# Y40001), changing media every day.

Mouse EpiSCs were established from ESC-46C after growth (4 weeks) in N2B27 basal medium containing 20 ng/ml of Activin (R&D, Cat# 338-AC-050) and 12 ng/ml FGF2 (Peprotech, Cat# 100-18B). The composition of the N2B27 basal medium was as follows: half of DMEM/F12 (Invitrogen, Cat# 21331-020), half of Neurobasal Medium (Invitrogen, Cat# 21103-049), 0.5× N2 (Invitrogen, Cat# 17502-048), 0.5× B27 (Invitrogen, Cat# 12587-010), 0.05 M β-mercaptoethanol (Invitrogen, Cat# 31350-010), and 2 mM L-glutamine (Invitrogen, Cat# 25030-024). EpiSCs were grown on Nunc plates coated with FCS (BioScience LifeSciences, Cat# 7.01 batch number 110006). Culture medium was changed every day and cells were split every other day, by washing 3 times with PBS without magnesium and calcium, incubating in PBS for 3 min at room temperature, gently scraping them from the plate pipetting up and down 3 times, before transferring to a new FCS-coated plate. The mouse EpiSC-Pitx3-GFP line was previously established in the Li laboratory (Jaeger *et al*, 2011), from ESC-Pitx3-GFP (Zhao *et al*, 2004). Frozen cell batches were tested for mycoplasma infection.

EpiSCs were differentiated into day 16 and 30 neurons with midbrain properties using a protocol optimized based on the method developed in Jaeger *et al* (2011).

The day before starting the differentiation protocol (day "−1"), growing EpiSCs were plated on Nunc plates coated with 15 μg/ml human plasma fibronectin (Millipore, Cat# FC010) and cultured in N2B27 basal medium containing Activin and FGF2, to reach 70–80% confluency after 24 h. Differentiation was started by rinsing cells twice with PBS, and culturing in N2B27 basal medium plus 1 μM PD 0325901 (Axon, Cat# 1408) for 2 days. Medium was refreshed every day. On day 2, cells were washed with PBS, scraped, replated on Nunc 10-cm dishes coated with 15 μg/ml human plasma fibronectin (Millipore, Cat# FC010), and cultured in N2B27 basal medium for 3 days. At this stage, medium was refreshed every day by removing half of the volume and adding half of the volume of freshly prepared medium.

After 72 h, the medium was replaced with N2B27 basal medium plus 100 ng/ml FGF8 (Peprotech, Cat# 100-25-25) and 200 ng/ml Shh (R&D, Cat# 464-sh-025). Medium was refreshed every day by removing half of the volume and adding half of the volume of freshly prepared medium.

After 96 h, cells were washed with PBS, and the medium replaced with N2B27 basal medium plus 10 ng/ml BDNF (R&D, Cat# 450-02-10), 10 ng/ml GDNF (R&D, Cat# 450-10-10), and 200 μM L-ascorbic acid (Sigma, Cat# A4544). Medium was refreshed every day by removing half of the volume and adding half of the volume of freshly prepared medium until days 16 and 30.

EZH1 and EZH2 enzymatic inhibition on ESCs, day 3, and day 30 differentiated cells was carried out adding 1 or 3 μM of UNC1999 (Konze *et al*, 2013; Sigma, Cat# SML0778) dissolved in DMSO, which were added directly to the medium. Cells treated with UNC1999 and control cells treated with DMSO were incubated for 24 h with drug or DMSO alone before RNA collection.

## mRNA expression

Total RNA was prepared from cells using TRIzol (Invitrogen, Cat# 15596-018) extraction following the manufacturer's instructions. Total RNA was treated with TURBO DNase I (Ambion, Cat#

AM1907) according to the manufacturer's instruction. Extracted RNA (1 μg) was reverse transcribed with 50 ng random primers and 10 U reverse transcriptase (Superscript II kit, Invitrogen, Cat# 11904-018) in a 20-μl reaction. The synthesized cDNA was diluted 1:10, and 2.5 μl used for qRT–PCR. Results were normalized to *Actb*. mRNA libraries were made using TruSeq RNA Sample

**Table 1. List of primers for qRT–PCR (5′–3′).**

| *Actb* | Fw | TCTTTGCAGCTCCTTCGTTG |
|---|---|---|
| | Rev | ACGATGGAGGGGAATACAGC |
| *Nanog* | Fw | ATGAAGTGCAAGCGGTGGCAGAAA |
| | Rev | CCTGGTGGAGTCACAGAGTAGTTC |
| *Oct4* | Fw | CTGAGGGCCAGGCAGGAGCACGAG |
| | Rev | CTGTAGGGAGGGCTTCGGGCACTT |
| *Fgf5* | Fw | CTTCTGCCTCCTCACCAGTC |
| | Rev | CACTCTCGGCCTGTCTTTTC |
| *Blbp* | Fw | GGGTAAGACCCGAGTTCCTC |
| | Rev | ATCACCACTTTGCCACCTTC |
| *Hes5* | Fw | AAGTACCGTGGCGGTGGAGAT |
| | Rev | CGCTGGAAGTGGTAAAGCAGC |
| *Mash1* | Fw | TGGAGACGCTGCGCTCGGC |
| | Rev | CGTTGCTTCAATGGAGGCAAATG |
| *Sox2* | Fw | CATGTGAGGGCTGGACTGCG |
| | Rev | GCTGTCGTTTCGCTGCGG |
| *Skor1* | Fw | GCTAAGTGCATCAAGTGCGG |
| | Rev | CACGAGTTAAAGTTGGCGGC |
| *Gata4* | Fw | GAGGCTCAGCCGCAGTTGCAG |
| | Rev | CGGCTAAAGAAGCCTAGTCCTTGCTT |
| *HoxA7* | Fw | AAGCCAGTTTCCGCATCTACC |
| | Rev | GTAGCGGTTGAAATGGAATTCC |
| *Pitx1* | Fw | CAGATCAGCGTCGGACGATT |
| | Rev | CACCACCACCGCACGAC |
| *Mesp1* | Fw | CGCAGTCGCTCGGTCC |
| | Rev | GGATGCTGTTTCTGCGTACAG |
| *Pou2f3* | Fw | GATCCTCTTTAGAGCCCCACCTG |
| | Rev | TCAGTGGGCTCATCATTCCCTC |
| *Pkp1* | Fw | TACGAATGCTTCCAGGACCA |
| | Rev | ACTTCTGCCGTTTGACGGT |
| *She* | Fw | CCGTAAGAACTCCCGGGTCA |
| | Rev | TCCCATTCTTCTCCTGCGTG |
| *Fam150b* | Fw | CGCTGCTGAGGTTGCTAGTT |
| | Rev | CTCTGCTCCTCTTGGTCTGC |
| *Fgf20* | Fw | GAGGCAGCAAGAAGTGCGA |
| | Rev | ATTTCAGTCATCCTGTCCGGC |
| *Chrdl2* | Fw | GAAAAGATATATACCCCCGGCCA |
| | Rev | TCAGAGCAGGTACAGCGCAC |
| *Tal1* | Fw | GCACCTAGTCCTGCTCAACG |
| | Rev | GGCCCTTTAAGTCCCTCGC |

Preparation Kits v2 setA (Illumina, Cat# RS-122-2001) and sequenced paired-end using Illumina Sequencing Technology by an Illumina HiSeq2000 following the manufacturer's instructions. The primers used are reported in Table 1.

### Incorporation of BrdU in replicating cells

To measure replication index, cells were grown in their normal medium in the presence of 50 μM BrdU (Sigma, Cat# 59-14-3) for 24 h. Cells were fixed (10 min) with 4% freshly depolymerized paraformaldehyde in 125 mM HEPES (pH 7.8). After washing (2×) in PBS, cells were permeabilized (20 min) with 0.2% Triton X-100 in PBS and blocked (30 min) with blocking solution (0.5% BSA, 0.2% fish skin gelatin in PBS). Cells were incubated in blocking solution (1 h, humid dark chamber) to which were added 0.5 U/μl DNase I (Sigma, Cat# D4263), 3 mM $MgCl_2$, and anti-BrdU antibody (1:150; raised in mouse which detected BrDNA after *in vivo* incorporation of BrdU into replicating DNA; Caltag Laboratories, Cat# MD5110). Cells were washed (3×, 3 min each) with blocking solution and incubated (40–60 min) with blocking solution containing the secondary antibody anti-Mouse IgG conjugated with Alexa Fluor 555 (1:1,000; raised in donkey; Invitrogen, Cat# A31570), in a humid dark chamber. Cells were rinsed (2×) with blocking solution and (2×) with PBS. Cells were stained (5 min) in 0.1 mg/ml DAPI in PBS, washed (1×) with PBS, and mounted using Dako Fluorescence Mounting Medium (Dako Agilent Technologies, Cat# S3023).

### Protein immunofluorescence

Cells were fixed (10 min) with 4% freshly depolymerized paraformaldehyde in 125 mM HEPES, washed twice with PBS, and permeabilized (3×, 5 min each) with 0.3% Triton X-100 in PBS (PBST), by gently rocking. Cells were blocked (30 min) with 3% donkey serum in PBST and then incubated (at 4°C, overnight), with primary antibody in PBST+3% donkey serum. Primary antibodies used were as follows: mouse Tuj1 (1:500; Sigma, Cat# T8660); mouse anti-OCT4 (1:50; BD Biosciences, Cat# 611202); sheep anti-TH (1:500; Pel-Freez, Rogers-Arkansas, Cat# P60101-0). After washing (3×, 10 min each) with PBST, cells were incubated (1 h, in a humid dark chamber) with secondary antibodies Alexa Fluor 555 anti-mouse IgG (1:1,000; raised in donkey; Invitrogen, Cat# A31570) or Alexa Fluor 555 anti-sheep IgG (1:500; raised in donkey; Invitrogen, Cat# A21436) in PBST+3% donkey serum. After three washes in PBS (10 min each), cells were incubated (5 min) in 0.1 mg/ml DAPI in PBS, washed with PBS (3×, 10 min each), before mounting in Dako Fluorescence Mounting Medium (Dako, Cat# S3023).

For double immunostaining of LMX1A and FOXA2, cells were permeabilized (3×, 5 min each) with 0.3% Triton X-100 in PBS (PBST), by gently rocking. Cells were blocked (30 min) with 3% donkey serum in PBST and then incubated (at 4°C, overnight), with anti-FOXA2 antibodies (1:200; raised in goat; Santa Cruz, Cat# sc-6554) in PBST+3% donkey serum. After washing (3×, 10 min each) with PBST, cells were incubated (1 h, in a humid dark chamber) with secondary antibodies Alexa Fluor 555 anti-goat IgG (1:500; raised in donkey; Invitrogen, Cat# A21432) in PBST+3% donkey serum. Cells were washed (3×, 10 min each) with PBST+3% donkey serum cells, stored (5 h) in 3% donkey serum in PBST, before incubation (at 4°C, overnight), with guinea pig LMX1A (1:500; clone GP6; custom-made antibody generated in the Li laboratory) in PBST+3% donkey serum. After washing (3×, 10 min each) with PBST, cells were incubated (1 h, in a humid dark chamber) with secondary antibodies anti-guinea pig IgG conjugated with Alexa Fluor 488 (1:500; raised in goat; Invitrogen, Cat# A11073) in PBST+3% donkey serum.

After three washes in PBS (10 min each), cells were incubated (5 min) in 0.1 mg/ml DAPI in PBS, washed with PBS (3×, 10 min each), before mounting in Dako Fluorescence Mounting Medium (Dako, Cat# S3023).

For triple immunostaining of dopaminergic, GABAergic, and glutamatergic markers (Fig EV3G), fixed cells were washed (3x) in PBST (PBS containing 0.1% Triton X-100) and incubated 2 h in a blocking solution that contained 5% donkey serum and 3% BSA in PBST. Primary antibodies used were as follows: sheep anti-TH (1:500, Pel-Freez, Rogers-Arkansas, Cat# P60101-0), mouse anti-VGluT2 (vesicular glutamate transporter 2, 1:1,000, Millipore, Cat# MAB5504), and rabbit anti-GABA (gamma-aminobutyric acid, 1:1,000, Sigma, Cat# A2052). The primary antibodies were diluted in blocking solution and incubated overnight at 4°C. The next day, cells were washed (3×) with PBST and bathed (1 h) in secondary antibodies Alexa Fluor 488 anti-sheep IgG (1:200, raised in donkey, Molecular Probes, Cat# A-11015), Alexa Fluor 555 anti-rabbit IgG (1:200, raised in donkey, Molecular Probes, Cat# A-31572), and Alexa Fluor 647 anti-mouse IgG (1:200, raised in donkey, Molecular Probes, Cat# A-31571) diluted in blocking solution, and after three washes in PBS, mounted with Dako Fluorescence Mounting Medium (Dako, Cat# S3023).

### Microscopy

Single immunofluorescence images (Fig 1C and D) were collected sequentially on a confocal laser-scanning microscope (Leica TCS SP8 STED; HC PLAPOCS 20×/0.75 IMM objective), equipped with a 405 diode and a WLL (Supercontinuum visible) laser, and pinhole equivalent to 1 Airy disk. For comparison of the different staining during neuronal differentiation, images were collected on the same day using the same settings, and without saturation of the intensity signal.

Double and triple immunofluorescence images (Figs EV1C and EV3G) were acquired on a confocal laser-scanning microscope (Leica TCS SP5; HCX PL APO CS 40.0× 1.25 OIL objective) equipped with a 405 nm diode, and Argon (488 nm), HeNe (543 nm), and HeNe (633 nm) lasers.

### Electrophysiology

Coverslips of mature neuronal cultures were placed onto a recording chamber and viewed using an Olympus BX51WI microscope with a 40× water immersion lens and DIC (differential interference contrast) optics. Cells were bathed at room temperature in a solution containing (in mM): 140 NaCl, 3.5 KCl, 1.25 $NaH_2PO_4$, 2 $CaCl_2$, 1 $MgCl_2$, 10 glucose, and 10 HEPES (pH 7.4). For whole-cell electrophysiological recordings, low resistance recording pipettes (8–14 MΩ) were pulled from capillary glass (Harvard Apparatus Ltd.) and coated with ski wax (Toko) to reduce pipette capacitance. Recording pipettes were filled with a solution containing: 140 mM K-gluconate, 5 mM NaCl, 2 mM ATP, 0.5 mM GTP, 0.1 mM $CaCl_2$,

1 mM $MgCl_2$, 1 mM ethylene glycol-bis (b-aminoethyl ether)-*N,N, N',N'*-tetraacetic acid (EGTA), and 10 mM HEPES (pH 7.4). For recordings of mIPSCs, a high-chloride solution was used: 135 mM CsCl, 4 mM NaCl, 2 mM ATP, 0.3 mM GTP, 0.5 mM $MgCl_2$, 2 mM EGTA, and 10 mM HEPES (pH 7.4). Pitx3-GFP positive neurons were identified and targeted using fluorescence via a GFP selective filter (X-Cite series 120, EXFO). For analysis of synaptic physiology, cells were held in voltage-clamp mode at $-70$ mV and events were detected over 3 min of recording. The first 30 s of recording was used for analysis of action potentials (i.e., frequency and CV-ISI). Cells were excluded from analysis if they exhibited a holding current $> -150$ pA, a series resistance $> 30$ M$\Omega$, or if the holding current did not remain stable over the course of recording. Antagonists were dissolved in the extracellular solution and applied to the cultures via bath perfusion at 30–33°C. The sodium channel blocker tetrodotoxin citrate (TTX) was used at 1 $\mu$M (Tocris Bioscience, Cat# 1069). In order to block glutamatergic synaptic activity, 2,3-dioxo-6-nitro-1,2,3,4-tetrahydrobenzo[f]quinoxaline-7-sulfonamide (NBQX; Sigma, Cat# N171), an AMPA receptor antagonist, was used at 5 $\mu$M, while Picrotoxin (100 $\mu$M, Sigma, Cat# P1675), a GABA$_A$ receptor antagonist, was used to identify GABAergic synaptic events. Data were acquired using an Axon Multiclamp 700B amplifier and a Digidata 1440a acquisition system, with pClamp 10 software (Molecular Devices). Data analysis was carried out using Clampfit 10.2 software (Axon Instruments), OriginPro 8.1 (OriginLab Corporation), Spike2v5 software (Cambridge Electronic Design), and MiniAnalysis software v6.0.7 (Synaptosoft). Data are presented as either percentage of cells or the mean $\pm$ standard error of the mean (SEM). Statistical analyses were performed using either chi-squared test for trend or a one-way ANOVA (Kruskal–Wallis; GraphPad Prism, GraphPad Software, San Diego, CA). Significance was noted when $P < 0.05$.

**Chromatin immunoprecipitation (ChIP), ChIP-seq, and re-ChIP**

The chromatin from ESCs or differentiated cultures was prepared by cross-linking in 1% formaldehyde for 10 min as previously described (Stock *et al*, 2007; Brookes *et al*, 2012). Arbitrary "chromatin concentration" was obtained by measuring absorbance (280 nm) of alkaline-lysed chromatin, and using the conversion 50 mg/ml for 1 absorbance unit. For mouse anti-RNAPII-S5p (clone CTD4H8, BioLegend, Cat# 904001), mouse anti-digoxigenin (Jackson Immunoresearch, Cat# 200-002-156), and rat anti-RNAPII-S7p (4E12; Chapman *et al*, 2007; kindly provided by Dirk Eick) antibodies, protein-G-magnetic beads (Active Motif, Cat# 53014) were incubated with rabbit anti-mouse (IgG + IgM) bridging antibodies (Jackson Immunoresearch, Cat# 315-005-048; 10 $\mu$g per 50-$\mu$l beads) for 1h at 4°C and washed with sonication buffer. For rabbit anti-H3K27me3 (Millipore, Cat# 07-449) antibodies, magnetic beads were just washed with sonication buffer. Fixed (700 $\mu$g) chromatin was immunoprecipitated (4°C, overnight) with the specific antibody and 50-$\mu$l beads (with/without bridging antibody), as previously described (Stock *et al*, 2007) and (Brookes *et al*, 2012). After immunoprecipitation using IgG antibodies, beads were washed as described previously (Stock *et al*, 2007). Immune complexes were eluted from beads (65°C, 5 min then room temperature, 15 min) with 50 mM Tris–HCl pH 8.0, 1 mM EDTA, and 1% SDS. Elution was repeated once and eluates pooled. Reverse cross-linking was

carried out (8 h, 65°C) after addition of NaCl and RNase A. EDTA was then increased to 5 mM, and samples were incubated (50°C, 2 h) with 200 $\mu$g/ml proteinase K (Roche, Cat# 3115836001). DNA was recovered by phenol-chloroform extraction and ethanol precipitation. The final DNA concentration was determined by Quant-iT PicoGreen dsDNA Assay Kit fluorimetry (Thermo Fisher Scientific, Cat# P7589). The quality of immunoprecipitated DNA used for high-throughput sequencing was confirmed prior to library preparation by quantitative real-time PCR (qPCR) analyses of Active, PRC/S5p, and Inactive genes with previously characterized chromatin states (Brookes *et al*, 2012); these quality control tests are not shown. Samples were diluted to the same concentration (0.2 ng/$\mu$l). The same amount of immunoprecipitated and input DNA (0.5 ng) was analyzed by qPCR using primers with the previously published sequences (Brookes *et al*, 2012). Amplifications (40 cycles) were performed using SensiMix No-ROX Kit (Bioline, Cat# QT650-05) with DNA Engine Opticon 1/2 Real-Time PCR system (Bio-Rad, Hemel Hempstead, Hertfordshire, UK).

ChIP-seq libraries were prepared from 10 ng DNA (quantified by PicoGreen and Qubit) using the Next ChIP-Seq Library Prep Master Mix Set for Illumina (NEB, Cat# E6240) following the NEB protocol, with some modifications. The intermediate products from the different steps of the NEB protocol were purified using MinElute PCR purification kit (Qiagen, Cat# 28004). Adaptors, PCR amplification primers, and indexing primers were from the Multiplexing Sample Preparation Oligonucleotide Kit (Illumina, Cat# PE-400-1001). Samples were PCR amplified prior to size selection on an agarose gel (250–600 bp including adapters). The size range was selected due to the broad distribution of RNAPII and H3K27me3 at promoters, where there is evidence that protection of DNA may occur (Ferrai *et al*, 2007, 2010). After purification by QIAquick Gel Extraction kit (Qiagen, Cat# 28704), libraries were quantified by Qubit (Invitrogen) and by qPCR using KAPA Library Quantification Universal Kit (KapaBiosystems, Cat# KK4824). Library size was assessed before high-throughput sequencing by Bioanalyzer (Agilent) using the High Sensitivity DNA analysis kit (Agilent, Cat# 5067-4626). Fragment sizes were within the selected size distribution. ChIP-seq libraries were sequenced single-end using Illumina Sequencing Technology using an Illumina HiSeq2000, according to the manufacturer's instructions.

Re-ChIP experiments were performed as previously described (Brookes *et al*, 2012). The first ChIP was performed as described above with the exception that after the final wash in TE buffer, immune complexes were eluted from the beads twice (65°C, 5 min, and room temperature, 15 min) using elution buffer (50 mM Tris pH 8.0, 1 mM EDTA, and 1% SDS). "Re-ChIP dilution" buffer (55 mM HEPES pH 7.9, 154 mM NaCl, 1.0 mM EDTA, 1.1% Triton X-100, 0.11% Na-deoxycholate) was added to dilute the eluate 10-fold to reach a final SDS concentration of 0.1%. The second round of immunoprecipitation was set up following the standard ChIP protocol. Enrichment relative to the input chromatin material was obtained using qPCR (see above) using the same amount of DNA in each PCR. To test carryover from the first immunoprecipitation, an unrelated antibody anti-digoxigenin (raised in mouse; Jackson Immunoresearch, Cat# 200-002-156) was used in the second round of immunoprecipitation. As a positive control, RNAPII-S5p ChIP was repeated in the second immunoprecipitation. The primers used for Re-ChIP are reported in Table 2.

**Table 2.   List of Re-ChIP primers (5′–3′).**

| Actb | Fw | GCAGGCCTAGTAACCGAGACA |
|---|---|---|
| | Rev | AGTTTTGGCGATGGGTGCT |
| Pitx1 | Fw | CAGATCAGCGTCGGACGATT |
| | Rev | CACCACCACCGCACGAC |
| Gata4 | Fw | TTCCCAGAAAACCGGCGCGA |
| | Rev | CCCTTAGGCCAGTCAGCGCA |
| Skap2 | Fw | TCCGGCTGTCCAGGGAGGAT |
| | Rev | CACCCTATGCGGACGGTGGG |
| Ajuba | Fw | GCCCTGCCTCTGCCTCTGTC |
| | Rev | CGAGTGACGGCTCTCCTGGC |
| Myf5 | Fw | GGAGATCCGTGCGTTAAGAATCC |
| | Rev | CGGTAGCAAGACATTAAAGTTCCGTA |

## Bioinformatics

### Read mapping and visualization

Sequenced single-end (50–51 bp) reads originating from ChIP-seq libraries were aligned to the Mouse reference genome mm10 (Dec. 2011, GRCm38/mm10) using Bowtie2 v2.0.5 (Langmead & Salzberg, 2012). The reference genome was indexed, and the alignments were performed with default parameters. Replicated reads (i.e., identical reads, aligned to the same genomic location), occurring more often than the 95th percentile of the frequency distribution of each dataset, were removed. When reads originated from multiple runs (technical replicates of sequencing), datasets were merged after mapping and removal of replicated reads.

Sequenced paired-end (2 × 100bp) reads from mRNA-seq libraries were aligned for visualization using TopHat v2.0.8 (Trapnell et al, 2009). The gene references provided to TopHat were the gtf annotation from UCSC Known Genes (mm10, version 6) and the associated isoform-gene relationship information from the Known Isoforms table (UCSC). Tables were downloaded from the UCSC Table browser (http://genome.ucsc.edu/cgi-bin/hgTables). UCSC Genome Browser (http://genome.ucsc.edu) was used for Figs 4B and 6E, and EV2A, and EV3G using default view settings, except: Windowing function—Mean, Smoothing Window—2 pixels and inclusion of zero on y-axis. Number of mapped reads for the datasets produced for this study is shown in Table 3.

### Non-redundant gene list selection

To investigate the most representative isoform for each gene in each time point, we followed the same strategy as in Brookes et al (2012), applied to mouse genome assembly mm10. To define isoforms and link them to gene clusters, we used UCSC Known Gene table (mm10, version 6) and the associated UCSC Known Isoforms table. Genes belonging to the mitochondrial genome (chrM) or "random chromosomes" (chr_random) were discarded. We also discarded clusters not linked to any RefSeq annotation (from the associated UCSC kgXref table). After filtering, we obtained a dataset of 23,177 gene clusters, associated with one or more RefSeq genes.

Around half of the gene clusters, 11,331, had a single isoform. When more than one isoform (UCSC Known Gene identifier) was present in a cluster, we selected one isoform per time point using

**Table 3.   List of genome-wide ChIP-seq and mRNA-seq datasets produced in the present study.**

| | Cell type | Read type | Mapped reads (millions) |
|---|---|---|---|
| ChIP-seq dataset | | | |
| RNAPII-S5p | 46C- ESC | 1 × 50 bp | 46 |
| RNAPII-S5p | 46C- day1 | 1 × 50 bp | 82 |
| RNAPII-S5p | 46C- day3 | 1 × 50 bp | 75 |
| RNAPII-S5p | 46C- day16 | 1 × 50 bp | 95 |
| RNAPII-S5p | 46C- day30 | 1 × 50 bp | 58 |
| RNAPII-S7p | 46C- ESC | 1 × 50 bp | 68 |
| RNAPII-S7p | 46C- day1 | 1 × 50 bp | 65 |
| RNAPII-S7p | 46C- day3 | 1 × 50 bp | 87 |
| RNAPII-S7p | 46C- day16 | 1 × 50 bp | 50 |
| RNAPII-S7p | 46C- day30 | 1 × 50 bp | 81 |
| H3K27me3 | 46C- ESC | 1 × 50 bp | 40 |
| H3K27me3 | 46C- day1 | 1 × 50 bp | 42 |
| H3K27me3 | 46C- day3 | 1 × 50 bp | 62 |
| H3K27me3 | 46C- day16 | 1 × 50 bp | 48 |
| H3K27me3 | 46C- day30 | 1 × 50 bp | 32 |
| Control (Dig) | 46C- ESC | 1 × 51 bp | 67 |
| Control (Dig) | 46C- day16 | 1 × 50 bp | 21 |
| polyA mRNA-seq dataset | | | |
| ESC | 46C- ESC | 2 × 100 bp | 48 |
| day1 | 46C- day1 | 2 × 100 bp | 46 |
| day3 | 46C- day3 | 2 × 100 bp | 47 |
| day16 | 46C- day16 | 2 × 100 bp | 40 |
| day30 | 46C- day30 | 2 × 100 bp | 51 |

the following criteria: (i) We selected the gene isoform with the highest amount of reads for RNAPII-S5p in a 2-kb window centered on the TSS (Transcription Start Site; in 15.7–17.1% of gene clusters the isoform was selected at this step); (ii) if ambiguity was still present, we selected the isoform with the highest RNAPII-S2p in a 4-kb window centered on the TES (Transcription End Site; in 13.3–13.9% of gene clusters the isoform was selected at this step); (iii) if ambiguity was still present, we selected the canonical isoform annotated in the associated UCSC mm10 known Canonical table (the gene with the highest number of coding bases; in 20.5–21.3% of gene clusters, the isoform was selected at this step); and (iv) if ambiguity was still present, one isoform was selected randomly (in 0.1–0.2% of gene clusters, the isoform was selected at this step).

### Promoter classification

To define genome-wide enriched regions for RNAPII-S5p, RNAPII-S7p, and H3K27me3, for each time point, we used Bayesian Change-point Model (BCP) peak-finder (Xing et al, 2012). BCP performs well in detecting enriched regions for broad chromatin features such as (broad) histone marks or RNAPII (Harmanci et al, 2014; Thomas et al, 2016). BCP was run in Histone Mark (HM) mode using as control dataset: (i) control (digoxigenin) ChIP-seq from ESCs, for

ESC, day 1, and day 3 datasets or (ii) control (digoxigenin) ChIP-seq from day 16 neurons, for day 16 and 30 datasets. Gene promoters were considered positive for RNAPII-S5p, RNAPII-S7p, or H3K27me3 when: (i) the 2-kb windows centered on the TSS overlapped with a region enriched for the mark, and (ii) the amount of reads in the TSS window was above a threshold. The threshold was defined as the 5th percentile of the distribution of reads in the TSS window of positive genes (5% tail cut).

To remove overlapping genes, we excluded genes whose (positive) TSS window overlapped positive windows for the same mark of other genes for more than 10% (200 bp). Positive genes that were inside other positive genes for the same mark were also excluded (internal gene removed). Excluded genes are marked as NA in Dataset EV2. Genes classified as NA for at least one of the three marks used for the classification (RNAPII-S5p, RNAPII-S7p, or H3K27me3) in one time point were excluded from further analyses. The genes that passed these filters (19,352) were classified, in each time point, according to the combination of RNAPII and H3K27me3 as belonging to one of eight different promoter states using the logics presented in main text (Fig 5A). The complete promoter classification is shown in Dataset EV2.

### mRNA-seq analyses

To calculate expression estimates, mRNA-seq reads were mapped with STAR (Spliced Transcripts Alignment to a Reference, v2.4.2a; Dobin *et al*, 2013) and processed with RSEM (RNA-Seq by Expectation-Maximization, v1.2.25; Li & Dewey, 2011). The references provided to STAR and RSEM were the gtf annotation from UCSC Known Genes (mm10, version 6) and the associated isoform-gene relationship information from the Known Isoforms table (UCSC). Both tables were downloaded from the UCSC Table browser (http://genome.ucsc.edu/cgi-bin/hgTables). Gene-level expression estimates in Transcript Per Million (TPM) were used for all the analyses and are reported in Dataset EV2.

To select genes that peak in one specific time point (Fig 1E), the expression levels from genes in the non-redundant list were standardized across time points using $z$-scores (calculated as the TPM value in each time point minus the mean value across the five time points, divided by the standard deviation). Then, genes whose standardized expression was higher than 1.75 in one time point were selected. Genes needed to be expressed (> 1 TPM) in at least one time point to be included in the analysis.

Statistical tests were performed using R, and the name of the test is specified in main text.

### Gene ontology enrichment

Gene ontology (GO) enrichment analysis was performed using GO-Elite version 1.2.5 (Gladstone Institutes; http://genmapp.org/go_elite). Pruned results (to decrease term redundancy) are reported. Default parameters were used as filters: $z$-score threshold more than 1.96, $P$-value < 0.05, number of genes changed more than 2. Over-representation analysis was performed with "permute $P$-value" option, 2,000 permutations. Permute $P$-value is reported in figures. UCSC Known Gene IDs were converted into the correspondent Ensembl Gene IDs (using the UCSC KnownToEnsembl table, downloaded from the UCSC Table browser http://genome.ucsc.edu/cgi-bin/hgTables) before performing the GO enrichment analyses. The group of genes used as background for each analysis is specified in

the associated figure legend. The complete list of enriched pruned GO terms is shown in Dataset EV1.

### Comparisons with published datasets

To compare PRC-positive genes in this study with previously published work, the list of genes whose promoter is classified as positive for H3K27me3 in ESC-46C was overlapped with available resources as follows: (i) List of genes positive for H3K27me3 in Young *et al*, 2011 defined using the classification provided in Supplementary Table 2 of the corresponding paper. In particular, genes classified as "Marked", "Broad", "Promoter", or "TSS" in the "ES H3K27me3" column were considered. Genes were matched with our classification using Ensembl Gene ID identifiers and converted from UCSC IDs as previously described. (ii) List of genes positive for H3K27me3 in Lienert *et al*, 2011 or in Brookes *et al*, 2012 were obtained from Table S2 in Brookes *et al*, 2012; that is, the genes classified as "1" in columns "H3K27me3 (TSS) (Lienert *et al*, 2011)" or "H3K27me3 (TSS) (Mikkelsen *et al*, 2007)", respectively. Genes were matched with our classification using Gene Symbols. Percentages were calculated as the number of genes positive in both classifications divided for the number of ESC-46C H3K27me3-positive genes present in the published lists.

To compare PRC/S5p genes in this study with bivalent genes, the list of genes whose promoter is classified as positive for H3K27me3 and RNAPII-S5p in ESC-46C was overlapped with list of bivalent genes ("K4+K27", "ESC") in Supplementary Table 3 in Mikkelsen *et al*, 2007. Genes were matched with our classification using Gene Symbols. We find 69% of bivalent genes in Mikkelsen *et al*, 2007 are positive for H3K27me3 and RNAPII-S5p in our ESC datasets.

### Upregulated genes in published Polycomb-knockout datasets

To explore the effect of PRC1/2 knockout in ESCs on the genes which are marked by H3K27me3 in presence or absence of RNAPII-S5p, we mined published resources. Expression values of wild-type and *Eed*-knockout ESCs (Cruz-Molina *et al*, 2017) were obtained from GEO Gene Expression Omnibus repository (GSE89210). Identifiers were then matched via Ensembl Gene IDs. Fold change was calculated as ratio between the average FPKM value in *Eed*-knockout ESCs relative to the average FPKM value in wild-type ESCs. Genes were classified as upregulated if the Fold change was > 2 and the expression in *Eed*-knockout ESC was > 1 FPKM. Genes classified as PRC/S5p, PRC Only or Inactive that were expressed > 1 FPKM in the published datasets for wild-type ESCs were excluded from the analysis, as these are unlikely to have matched repressed promoter states. The percentage of upregulated genes in each promoter state was calculated relative to the number of genes in that promoter state found in the published dataset.

Expression values of control cells and tamoxifen-induced conditional *Ring1B* knockout in *Ring1A*$^{-/-}$ knockout cells measured by microarray (Endoh *et al*, 2008) were obtained from GEO (GSM265042 and GSM265043) and analyzed as in (Brookes *et al*, 2012). Identifiers were matched via Gene Symbols. The percentage of genes upregulated in each promoter state was calculated in relation to the number of genes with the same promoter state that are present in the published datasets.

To explore the effect of PRC2 knockout in neurons on genes marked by H3K27me3 in the presence or absence of RNAPII-S5p, we mined a published list of upregulated genes from *ex vivo*

medium spiny neurons from 6 month old control and *Ezh1*$^{-/-}$; *Ezh2*$^{fl/fl}$; *Camk2a-cre* knockout mice, measured by microarray (from the Supplementary Table 3 in von Schimmelmann *et al*, 2016). Genes in the published list were matched with our day 30 list of promoter states using Gene Symbols. Microarray probes matching to more than one Gene Symbol were not considered. In the absence of the total list of detected genes, the percentage of genes upregulated in each promoter state was calculated relative to the total number of genes in that group.

### Plot generation

Heatmaps in Figs 1E and 3B, and EV4B were produced using the CRAN package *pheatmap* (Kolde, 2015) in R.

Bar plots, dot plots, and boxplots were produced using Excel or R. For mRNA-seq, a pseudo-count of $10^{-4}$ was added to TPM prior to logarithmic transformation and plotting. In Fig 7D, a pseudo-count of 1 was added to peak size H3K27me3 prior to logarithmic transformation and plotting.

To facilitate comparison between days, in Fig 3B, the number of reads mapping in 2-kb windows around the TSS (transformed in log scale) for each gene was divided by the threshold of minimal number of reads in positive genes (5% Threshold, see section on Promoter Classification). To help visualization, these values were then scaled to have the same maximum value, and the minimum and maximum values of the color scale were set to 1 and 99% of the distribution, respectively.

Average ChIP-seq profiles (Fig 7B and C) were generated as previously (Brookes *et al*, 2012), by plotting the average coverage in non-overlapping windows of 10 bp, across genomic windows centered on the TSS and the TES using custom scripts.

### Peak size, GC content, and CpG coverage calculation

To calculate the breath of H3K27me3 windows covering each gene (Fig 7D), we calculated the size of the BCP-detected enriched region that overlaps the 2-kb window centered on the TSS of each gene. If more than one peak was present, the longest was selected. To calculate the overlap between TSS window or gene body (TSS to TES) with CpG islands, we used the CpG island definition from the mm10 cpgIslandExt table, downloaded from UCSC Table Browser. For each gene, we used the isoform selected in day 30 neurons. The overlap was computed with IntersectBed, part of Bedtools v2.17.0 (Quinlan & Hall, 2010). To calculate the proportion of TSS window composed of G or C nucleotides, the sequence of the 2-kb window centered on the TSS used in day 30 was extracted using bedtools 2.17.0 getfast and analyzed using a custom script. Statistical tests were performed using R, and the name of the test is specified in the text.

### DNA hypomethylated region and DNA methylation valley definition

Hypomethylated regions (HMRs) from mouse brain, Frontal Cortex (Lister *et al*, 2013); dataset name: MouseFrontCortexMale22Mo), were downloaded from Methbase (Song *et al*, 2013); http://smithlabresearch.org/software/methbase), a resource from the Smith laboratory, accessed through the UCSC Table Browser. HMRs were downloaded as a bed file, already mapped to mm10. DNA methylation valleys (DMVs) were defined starting from HMR using custom scripts for proximity clustering and 5-kb size cut-off, following a previously described approach (Xie *et al*, 2013).

### Mogrify

To identify candidate TFs required to convert a given cell type into another regardless of their original identity, we extracted the influence ranks provided by the Mogrify algorithm (Rackham *et al*, 2016). These ranks are calculated for each TF for a conversion between any of 173 cell types or 134 tissue types based on the TFs estimated ability to regulate the differential expression of its immediate network neighborhood. For any given cell type, we take the average rank of the TF over every possible conversion; for instance, the TF Sox2 is often ranked highly for cell conversions into human neurons and as such has an average rank of 34.86 (the maximum possible rank is 303). The full list of genes and extracted average ranks for the transitions investigated here can be found in Dataset EV3.

Each human gene was then matched with the mouse homolog using homology table from Ensembl, release 84 (downloaded through BioMart http://www.ensembl.org/biomart/). Matching was performed on Gene Names. Enrichment or depletion in particular promoter states from day 30 neurons was tested in R with hypergeometric test.

Promoter state enrichment in genes with high or low influence on trans-differentiation (estimated with their rank) was tested using Gene Set Enrichment Analysis (Subramanian *et al*, 2005), in "GSEAPreranked" mode. GSEA tests whether a ranked list (here, the TFs involved in trans-differentiation) is enriched in certain Gene Sets (here, different promoter states in day 30 or all days). In "GSEAPreranked" mode, GSEA is run against a user-supplied, ranked list of genes (the list of TFs involved in trans-differentiation, using the influence ranks calculated by Mogrify) and tests for statistically significant enrichment at either end of the ranking. Enrichment was considered significant at a false discovery rate < 25%.

### Data availability

ChIP-seq and mRNA-seq data have been submitted to the GEO repository under accession number GSE94364.

- List of Enriched Gene Ontology (GO) terms and associated genes: Dataset EV1.
- List of Promoter classification and expression for genes in each time point: Dataset EV2.
- List of Mogrify genes: Dataset EV3.

**Expanded View** for this article is available online.

### Acknowledgements

We thank Laurence Game and Adam Giess (MRC LMS Sequencing Facility) for sequencing and raw data processing, Elsa Abranches, Domingos Henrique and Ines Jaeger for neuronal differentiation advice, and Dirk Eick for the RNAPII-S7p antibody. We thank the Pombo laboratory for helpful discussions and comments on the manuscript. CF and AP thank the BBSRC (UK); CF, ETT, TR, AK, AA, and AP thank the Helmholtz Association (Germany) for support. CF was recipient of a Wellcome Trust VIP award. This work was supported by the UK Medical Research Council (MRC; MC_U120061476) to AP; by the MRC (G117/560, U120005004) and EU framework program 7 (NeuroStemcell, no. 222943) to ML; by the MRC (U120085816) and a Royal Society University Research Fellowship to MAU. ML was an MRC Senior Non-Clinical Research Fellow. MN thanks CINECA ISCRA HP10CYFPS5 and HP10CRTY8P, computer resources at

INFN and Scope at the University of Naples, and the Einstein BIH Fellowship Award.

## Author contributions

CF and AP conceived and designed the project; CF, JRR-J, and AK conducted the experiments; ETT, TR, OJLR, IS, and AA designed and conducted bioinformatic analysis; MN helped with data analysis; ML supported the development of the neuronal differentiation protocol; MAU supervised the electrophysiological experiments; CF, ETT, JRR-J, TR, OJLR, MAU, and AP analyzed and interpreted the results; CF, ETT, MAU, and AP prepared the manuscript with the help of the other authors.

## Conflict of interest

The authors declare that they have no conflict of interest.

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
