## [Review Process File · Molecular Systems Biology]

RNA polymerase II primes Polycomb-repressed developmental genes throughout terminal neuronal differentiation

Carmelo Ferrai, Elena Torlai Triglia, Jessica R. Risner-Janiczek, Tiago Rito, Owen J.L. Rackham, Ines de Santiago, Alexander Kukalev, Mario Nicodemi, Altuna Akalin, Meng Li, Mark A. Ungless, Ana Pombo

Corresponding authors: Ana Pombo & Carmelo Ferrai, Berlin Institute for Medical Systems Biology and Mark Ungless, MRC London Institute of Medical Science, Imperial College

Review timeline:

Submission date:	20 May 2017
Editorial Decision:	09 June 2017
Revision received:	16 August 2017
Editorial Decision:	04 September 2017
Revision received:	08 September 2017
Accepted:	11 September 2017

Editor: Maria Polychronidou

Transaction Report:

1st Editorial Decision

09 June 2017

Thank you again for submitting your work to Molecular Systems Biology. We have now heard back from the two referees who agreed to evaluate your study. As you will see below, the reviewers think that the findings seem interesting and they acknowledge the quality of the data. They raise however some (mostly minor) concerns, which we would ask you to address in a revision of the manuscript.

I think that the reviewers' recommendations are rather clear but of course, feel free to contact me in case you would like to discuss any particular point in further detail.

REFeree REPORTS

Reviewer #1:

Ferrai et al. established an in vitro differentiation system that yields functional dopaminergic neurons and used it to probe the genome-wide changes of poised chromatin (Ser5 RNAPII and H3K27me3). They find that poised chromatin is not characteristic for embryonic stem cells but is also present in differentiated cells. Particularly developmental transcription factors, which are often CpG rich and have large H3K27me3 domains with co-occurring Ser5 RNAPII, remain poised in

differentiating cells, although their function is not needed.

Overall, this study is rigorously designed, comprehensively analyzed and well written and presented. Especially the development, characterization and genomics analyses of the *in vitro* differentiation system of dopaminergic neurons is nicely done and a great resource. I strongly support publication. In addition to very minor points (see below), I have one recommendation for the revision.

"Poised" chromatin has been extensively debated in the last 10 years. To this date, there is no clear consensus in the field which co-occurrence of activating and repressive marks represents a poised state and what functional implications a poised state entails. Is it a state that is more easily or more rapidly induced in response to a signal? Does the state indicate a preparation for future events or does it represent an abstract potential for regulation? Or is it simply a state that points to extensive regulation in development? I understand that the authors cannot clarify and discuss this in detail but I do think the authors can elaborate a bit more in their discussions what their results mean. They say that "Future studies ... will help determine...whether Polycomb repressive mechanisms in the presence of poised RNAPII complexes are linked to the potential of one cell type to dedifferentiation and/or selectively reprogram into another." This statement goes along with the assumption that the poised state has a purpose. However, there are other explanations for the perseverance of the poised state in differentiated cells. Evidence suggests that poised genes are induced more frequently during development and that regulation at these promoters predicts this (e.g. Gaertner et al. Cell Reports 2012) but it unclear whether a poised state really represents a mechanistic advantage since genes can be induced very rapidly without being in a poised promoter state beforehand (e.g. Lin et al. Genes Dev 2011). The fact that the poised promoter state is associated with distinct sequence features suggests an alternative, namely that highly regulated genes have evolved more elaborate mechanisms of regulation, which includes both repressive and activating features. In fact, this hypothesis fits the finding of the paper. If the poised state is a more intrinsic regulatory feature of the gene, rather than a state with biological purpose (e.g. dedifferentiation), then one might have expected that the poised state does not completely disappear in differentiated cells, even if the genes that have it are no longer needed in that cell type (and by the way, the perseverance of the poised state in differentiated cells has also been observed in *Drosophila*, see Gaertner et al. Cell Reports 2012). This regulatory hypothesis is likely true for Polycomb regulation, which is particularly extensive at highly regulated developmental genes such as transcription factors, and which is coupled to intrinsic sequence features (Polycomb response elements and CpG enrichment). In this model, Polycomb simply reinforces repression if transcription is weak. For this reason, I disagree with the statement at the end of the paper, when describing the depression at poised genes in *Ezh1/2* mutants, that "These observations suggest that PRC/S5p state of poising is associated with increased potential for reactivation." The observation only confirms that the poised genes are repressed by Polycomb, not that they have increased potential for reactivation in response to extracellular signals. This also questions the title of the paper, which emphasizes the role of RNAPII in "priming" promoters (why another term here?). I understand what the authors are trying to say and I don't disagree but I wish they would consider other interpretations of their data and demystify the term "poised".

Minor points:

Page 17: it says "vast majority" for 65%, which I find an exaggeration. I suggest changing it to "majority".

Figure 5d: The y axis should be labelled "Transcript levels [$\log_{10}(\text{TPM} + 0.0001)$]", otherwise it is not clear what is measured. And maybe the flooring to 0.0001 could be moved to the methods.

Figure 6d is a bit small. In my opinion, the axis can be cut and stretched to enlarge the data.

Reviewer #2:

I read with great interest the manuscript by Ferrai et al. entitled "RNA polymerase II primes polycomb-repressed developmental genes throughout terminal neuronal differentiation". In this manuscript, the authors extend their previous observations in ESC to the whole process of differentiation towards dopaminergic neurons and report that a significant fraction of polycomb-

repressed genes display poised RNAPII at their promoters at all analysed stages. Overall, the presented data seems to be of high quality and can be considered as a valuable resource to better understand neuronal differentiation. However, my main concern with this manuscript is that it is largely incremental with respect to previous work from the same laboratory and largely based on correlative observations. It would be nice if the authors make an attempt to investigate the biological relevance of having S5p RNAPII at the promoter regions of RNAPII:

- Are PRC/S5P genes easier to activate than PRC-only genes?. They could use dCas9-activator fusions to trans-activate candidate genes and test this idea.
- Upon loss of PRC2 in ESC (or during differentiation), are PRC/S5P genes more likely to become abnormally induced than PRC-only genes?. How is such induction correlated to the initial levels of S5P at the promoter regions?.

On the other hand, the authors should discuss their current and previous findings with respect to recent reports that challenge the poisoning of PRC genes by RNAPII (Williams et al, Mol Cell, 2015; Min et al., Genes Dev, 2011; Riising et al., Mol Cell, 2014). In these papers the authors report that most PRC genes in mESC display low or undetectable levels of poised RNAPII using GRO-seq or total RNAPII ChIP-seq. Moreover, they also showed that PRC2 is not necessary to maintain PRC-genes in a silent state in mESC. Can the authors generate GRO-seq or total RNAPII ChIP-seq to test their findings based on S5P RNAPII ChIP-seq?. Alternatively, they could at least use publically available data in mESC.

1st Revision - authors' response

16 August 2017

Point-by-point response

Reviewer #1:

“...Overall, this study is rigorously designed, comprehensively analyzed and well written and presented. Especially the development, characterization and genomics analyses of the in vitro differentiation system of dopaminergic neurons is nicely done and a great resource. I strongly support publication. In addition to very minor points (see below), I have one recommendation for the revision.”

"Poised" chromatin has been extensively debated in the last 10 years. To this date, there is no clear consensus in the field which co-occurrence of activating and repressive marks represents a poised state and what functional implications a poised state entails. Is it a state that is more easily or more rapidly induced in response to a signal? Does the state indicate a preparation for future events or does it represent an abstract potential for regulation? Or is it simply a state that points to extensive regulation in development? I understand that the authors cannot clarify and discuss this in detail but I do think the authors can elaborate a bit more in their discussions what their results mean.

We are grateful for the reviewer's words of support and share the concerns with aspects of the literature that currently confound our understanding of poised chromatin states, and more specifically the different mechanisms of RNAPII regulation across different biological systems. We welcome the reviewer's suggestion to expand this part of the manuscript (see revised text in p. 4 and 5).

Some aspects of the current 'confusion' relate in part with semantics, and the words Paused, Stalled, Primed and Poised which are used by different authors to describe different states of RNAPII activation/regulation in different contexts. We were the first to introduce the terminology "poised RNAPII" to specifically describe the unique RNAPII state which we identified at Polycomb repressed genes in ESCs (Stock et al. 2007 Nat Cell Biol; see Brookes and Pombo, 2009 EMBO Reports). This terminology was adopted by the Reinberg lab (Tee et al. 2014 Cell). The justification for a new nomenclature was that the 'poised' RNAPII state found in ESCs at Polycomb repressed chromatin was (and remains to this date) distinct from any other known RNAPII states (paused, stalled or active) described across different biological systems.

We have expanded the Introduction to include a more explicit statement of the unique properties of “poised RNAPII” (page 5):

The poised RNAPII is characterized by exclusive phosphorylation of S5p in the absence of S7p, S2p, K7me1/2, K7ac, or recognition by 8WG16 (Brookes et al, 2012; Dias et al, 2015). Poised RNAPII-S5p, in the absence of 8WG16, has not been described in Drosophila (Gaertner et al, 2012), consistent with lack of chromatin bivalency (Voigt et al, 2012, Vastenhouw and Schier, 2012). Importantly, Ser5 phosphorylation of poised RNAPII complexes at Polycomb repressed genes in ESCs is mediated by different kinases, ERK1/2 (Ma et al, 2016; Tee et al, 2014), instead of CDK7 which phosphorylates both S5p and S7p at active genes, irrespectively of pausing ratio (Akhtar et al, 2009; Glover-Cutter et al, 2009).

By calling these polymerases ‘poised’, we aimed to state that RNAPII is already present at specific promoters in ESCs, many days and hours before their potential (yet optional) activation during cell commitment. The terminology ‘poised RNAPII’ was not originally used to mean that the genes will inevitably be activated, as we know that some poised genes will never be activated in specific ESCs depending on the lineage that each cell gives rise to. If a subpopulation of cells in an ESC culture commits to neurons, the poising of RNAPII at cardiac developmental TFs would have served no functional or useful purpose in that subpopulation. Therefore, the poised state in a given time point of the cell’s life reflects potential for activation whether or not activation is ever fulfilled. In our view these genes are primed and to avoid further confusion, we would like to keep the nomenclature poised RNAPII-S5p to describe the state specifically found at PRC genes. We hope that this now better clarifies our views, and that our revised Introduction improves the understanding of the nomenclature and the previous work.

“They say that "Future studies ... will help determine...whether Polycomb repressive mechanisms in the presence of poised RNAPII complexes are linked to the potential of one cell type to dedifferentiation and/or selectively reprogram into another." This statement goes along with the assumption that the poised state has a purpose.”

Fair point and we have revised the sentence to (page 25):

“Future studies in other terminally differentiated cell types will help understand whether the poised chromatin state is an inherent state of repression for these important transcription factors, or alternatively whether distinct sets of poised PRC/S5p genes are specifically chosen in each specific cell type to restrict the potential to selectively reprogramming into another cell type, either during the normal physiology of a tissue or in disease.”

“However, there are other explanations for the perseverance of the poised state in differentiated cells. Evidence suggests that poised genes are induced more frequently during development and that regulation at these promoters predicts this (e.g. Gaertner et al. Cell Reports 2012) but it unclear whether a poised state really represents a mechanistic advantage since genes can be induced very rapidly without being in a poised promoter state beforehand (e.g. Lin et al. Genes Dev 2011).”

The reviewer refers to two studies where the S5p state specific of poised RNAPII complexes found at PRC-repressed genes was not directly investigated.

Lin et al. 2011 refers to *Cyp26a1*, a specific gene which is induced rapidly and without previous RNAPII occupancy (their Fig 6). When we investigate the state of RNAPII phosphorylation of the *Cyp26a1* gene in our current and published datasets (Brookes et al, 2012), we find that the *Cyp26a1* gene is clearly occupied by poised RNAPII-S5p irrespective of ESC line and growth conditions, above mock ChIP (Rebuttal Figure 1). Therefore, we argue that the quick reactivation described in Lin et al 2011 may, at least in part, be due to the presence of poised RNAPII complexes that were not detected due to choice of RNAPII antibodies. We confirm that *Cyp26a1* is not occupied by 8WG16, as shown in Lin et al. 2011, nor S7p or S2p.

Rebuttal Figure 1. ChIP-seq profiles of the *Cyp26a1* gene from two different ESC lines: clone OS25 grown in serum (Brookes et al. 2012) and clone 46C grown in serum-free conditions (present study). The *Cyp26a1* gene is classified as PRC/S5p as shown by its enrichment for RNAPII-S5p, H3K27me3 and H2Aub1, background levels of 8WG16, RNAPII-S7p and RNAPII-S2p, similar to mock ChIP levels, and little detectable mRNA. H3K27me3-Mikkelsen et al data was remapped from the published dataset from Mikkelsen et al. (2007).

Gaertner et al (2012) mostly focused on the study of ‘poised states’ in *Drosophila*, with a smaller investigation of H3K27me3 and RNAPII-8WG16 occupancy in ESCs.

In *Drosophila*, they investigated genes that are marked by H3K27me3 and RNAPII-8WG16 and not expressed (which they call ‘*balanced genes*’). To connect our study with Gaertner, we have clarified this point (page 5):

“Poised RNAPII-S5p, in the absence of 8WG16, has not been described in Drosophila (Gaertner et al. 2012), consistent with lack of chromatin bivalency (Voigt et al, 2012, Vastenhouw and Schier, 2012). Importantly, Ser5 phosphorylation of poised RNAPII complexes at Polycomb repressed genes in ESCs is mediated by different kinases, ERK1/2 (Ma et al, 2016; Tee et al, 2014), instead of CDK7 which phosphorylates both S5p and S7p at active genes, irrespectively of pausing ratio (Akhtar et al, 2009; Glover-Cutter et al, 2009).

Gaertner et al (2012) also define a group of *Drosophila* genes which are not marked by K27me3 but they called ‘poised’ based on the presence of 8WG16 and no expression, which we could interpret as a strictly paused state, such as it would be the case for an heatshock gene. They also define another group of genes not marked by K27me3 which they call ‘paused genes’ and are expressed and associated with presence of 8WG16. These states reflect important features of gene activation, but they are not marked by K27me3,

and are not the focus of our current manuscript.

Gaertner et al (2012) also perform a brief analysis of genes marked by 8WG16 in mouse ESCs to investigate their potential for activation using mRNA-seq datasets during a tight time course upon retinoic acid treatment to promote differentiation (their Fig 7). They were interested in specific H3K27me3⁺ promoter states (which are not called poised in their study):

- 775 'H3K27me3' genes, of these
- 445 'balanced' genes are associated with both 8WG16 and K27me3, and
- 483 'bivalent' genes, marked by H3K4me3 and H3K27me3 (most, 340 of bivalent genes overlap with the balanced group).

It is very difficult to integrate the analyses of Gaertner with our work, as Gaertner's K27me3 classification does not include many genes known to be under Polycomb repression in ESCs, probably because the published K27me3 dataset that they analysed is quite noisy (which we also mined in Brookes et al. 2012; Fig 4A therein). Out of approximately 2700 bivalent genes originally identified by Mikkelsen et al. (2007) we match 2352 in our promoter lists. We confirm the H3K27me3⁺ state of 2053 of these genes in our ESC dataset. We also confirm that most are S5p positive (1634), as shown in Brookes et al. (2012). In contrast, Gaertner identifies only 775 K27me3⁺ genes, which excludes prototypical developmental transcription factors such as Gata4 and Nkx2-9. These genes have been extensively found associated with PRC1 and PRC2 enzymes and their histone marks in ESCs, and with poised RNAPII-S5p and H3K4me3. Their state has been characterized in detail by single gene ChIP-qPCR in Azuara et al 2006, Stock et al 2007 (Fig 1A, Fig 5, S4), and Brookes et al. 2012 (SI Fig S3C).

Irrespective of the overlap between the three groups, Gaertner et al find that the H3K27me3 promoter state is less (or not) predictive for early gene activation within the first ours of RE treatment until 16-24h, in comparison with the other groups of genes identified as K27me3+K4me3⁺ or K27me3+8WG16⁺. To understand the different behaviors of these highly overlapping groups of genes and assess their association with S5p, we asked whether the K27me3⁺ that are negative for K4me3 in Gaertner (255 genes) are positive for RNAPII-S5p. We found most genes (184 out of 255) in this category. And we find that most of the Gaertner K27me3⁺ genes are positive for RNAPII-S5p⁺. Therefore, if we consider the analyses in Gaertner, it appears that the presence of poised RNAPII-S5p at K27me3⁺ genes may be associated with the delayed response to retinoic acid treatment.

Due to the significant differences in criteria for promoter classification and also in criteria of detectability of promoter features between these different studies, we have not included the comparisons described above in our current manuscript. We hope that the reviewer may agree that this topic is extremely complex, especially as different studies use different RNAPII phosphorylation states to classify promoters, and there is a much needed effort for integrated analyses and an extended review of the literature, which goes beyond the format of our current primary research manuscript.

“The fact that the poised promoter state is associated with distinct sequence features suggests an alternative, namely that highly regulated genes have evolved more elaborate mechanisms of regulation, which includes both repressive and activating features. In fact, this hypothesis fits the finding of the paper. If the poised state is a more intrinsic regulatory feature of the gene, rather than a state with biological purpose (e.g. dedifferentiation), then one might have expected that the poised state does not completely disappear in differentiated cells, even if the genes that have it are no longer needed in that cell type (and by the way, the perseverance of the poised state in differentiated cells has also been observed in Drosophila, see Gaertner et al. Cell Reports 2012). This regulatory hypothesis is likely true for Polycomb regulation, which is particularly extensive at highly regulated developmental genes such as transcription factors, and which is coupled to intrinsic sequence features (Polycomb response elements and CpG enrichment).”

We agree with the interpretation that the PRC/S5p promoter state is the chosen mechanism of repression for these important high CpG developmental genes, which might have emerged during evolution as the chosen mode of repression in the absence of DNA methylation. Nevertheless, the PRC/S5p promoter state may have roles in adult tissue physiology, for example during tissue regeneration, an area where Polycomb repression requires further investigation, and for which the concepts presented in our study will be an important resource.

To begin addressing a role for the poised RNAPII-S5p state at Polycomb repressed genes in differentiated cells, we explored Mogrify, an algorithm that identifies genes that are likely to drive trans-differentiation when overexpressed (Rackham et al., 2016, Nat Genet) and investigated whether the PRC/S5p state is a feature of the genes identified to have ‘high influence’ on transdifferentiation. Genes identified by Mogrify’s algorithm confirmed genes previously shown to induce transdifferentiation and made new predictions which were successfully validated experimentally (Rackham et al. 2016). Interestingly, amongst 11 genes used by Rackham to drive epidermal or endothelial transdifferentiation, three are classified as always PRC/S5p throughout our neuronal differentiation, seven are marked by poised RNAPII-S5p and Polycomb in ESCs, and one acquires Polycomb in day 3.

We present these new results in two additional figures Fig 8 and Fig EV7 and expanded the manuscript accordingly. We show that PRC/S5p genes are enriched among the genes that Mogrify identifies as highest influence to promote trans-differentiation. We feel that this additional integration of our datasets with the Mogrify algorithm may serve as a rich resource to link poised PRC/S5p genes with genes important for cell plasticity and in transdifferentiation strategies, which are aspects relevant for the development of adult cell therapies. We find this new section of the manuscript a valuable contribution, but would welcome the reviewers’ and editor’s advice about its value to strengthen our manuscript.

“In this model, Polycomb simply reinforces repression if transcription is weak. For this reason, I disagree with the statement at the end of the paper, when describing the depression at poised genes in Ezh1/2 mutants, that “These observations suggest that PRC/S5p state of poising is associated with increased potential for reactivation.” The observation only confirms that the poised genes are repressed by Polycomb, not that they have increased potential for reactivation in response to extracellular signals.

We understand the concerns of the reviewer, also highlighted by reviewer 2. To begin address this aspect, we have compared the potential for reactivation of PRC/S5p and PRC Only genes. We also expanded our analyses of Polycomb knockout experiments by analysing additional published datasets. Finally, we performed Polycomb inhibition in our cultures followed by single gene expression analyses. We have added an additional figure to our revised manuscript (Fig 6). We took two approaches:

1) We analyzed published datasets from ESCs and in vivo Neurons upon knockout of PRC1 or PRC2 subunits and compared the potential for reactivation of H3K27me3 that are associated with S5p (PRC/S5p genes) or not (PRC Only genes). In the new Fig 6A and B, we show that the PRC/S5p genes are more likely to become re-activated upon PRC knockout, in comparison with PRC Only genes, suggesting that the poised RNAPII-S5p confers increased potential for reactivation to Polycomb-repressed genes. The new results section is found on page 18.

2) We also inhibited PRC2 in our ESC and neuronal cultures (days 3 and 30) using a drug that blocks the enzymatic activity of both Ezh1 and Ezh2 (UNC1999, Konze et al, 2013). We find that PRC/S5p genes become derepressed within 24h of treatment, in contrast with PRC Only genes, indicating that RNAPII occupancy increases the potential for reactivation when Polycomb inhibition is interfered with (Fig. 6C). These results are presented and discussed in page 19.

Further work will be necessary to fully answer the important points raised by the reviewer. We hope that our current manuscript will revive the interest in understanding Polycomb repression mechanisms and inspire future analyses of RNAPII states in that repressive context.

This also questions the title of the paper, which emphasizes the role of RNAPII in "priming" promoters (why another term here?). I understand what the authors are trying to say and I don't disagree but I wish they would consider other interpretations of their data and demystify the term "poised".

We choose the word 'primed' as a generic term to refer to the presence of RNAPII at gene promoters, instead of the word 'poised' which in the literature is used to refer to many different RNAPII states and for that reason we feel would be more confusing. In the introduction, we now clarify that genes can be 'primed' in different ways by referring to both the 'paused' and 'poised' states, and more explicitly stating the features of the poised RNAPII-S5p state found at PRC-repressed genes.

"Minor points:

Page 17: it says "vast majority" for 65%, which I find an exaggeration. I suggest changing it to "majority".

We thank the reviewer for this suggestion, and have changed the sentence accordingly.

"Figure 5d: The y axis should be labelled "Transcript levels [log10 (TPM + 0.0001)]", otherwise it is not clear what is measured. And maybe the flooring to 0.0001 could be moved to the methods."

We changed the labeling on the Y axis and confirm that the flooring was explained in the Methods.

"Figure 6d is a bit small. In my opinion, the axis can be cut and stretched to enlarge the data."

We modified the panel (now referred as Fig 7D) as suggested.

Reviewer #2:

"I read with great interest the manuscript by Ferrai et al. entitled "RNA polymerase II primes polycomb-repressed developmental genes throughout terminal neuronal differentiation". In this manuscript, the authors extend their previous observations in ESC to the whole process of differentiation towards dopaminergic neurons and report that a significant fraction of polycomb-repressed genes display poised RNAPII at their promoters at all analysed stages. Overall, the presented data seems to be of high quality and can be considered as a valuable resource to better understand neuronal differentiation. However, my main concern with this manuscript is that it is largely incremental with respect to previous work from the same laboratory and largely based on correlative observations. It would be nice if the authors make an attempt to investigate the biological relevance of having S5p RNAPII at the promoter regions of RNAPII:

- Are PRC/S5P genes easier to activate than PRC-only genes? They could use dCas9-activator fusions to trans-activate candidate genes and test this idea.

- Upon loss of PRC2 in ESC (or during differentiation), are PRC/S5P genes more likely to become abnormally induced than PRC-only genes? How is such induction correlated to the initial levels of S5P at the promoter regions?"

The reviewer requested that we make an attempt to investigate the biological relevance of having S5p RNAPII at the promoter regions of Polycomb-repressed genes. Accordingly, we have taken two independent approaches to investigate whether the presence of RNAPII-S5p at PRC-repressed genes makes them more likely to be induced upon inhibition of PRC2 activity, as specifically suggested by the reviewer.

1) We analyzed published datasets from Polycomb knockouts both *using* ESCs and *in vivo* Neurons. The results that are now included in the new Fig 6A and B and they show that PRC/S5p genes are indeed more likely to be reactivated, as the reviewer predicted might be the case.

2) We treated ESCs, day 3 and day 30 neurons, with UNC1999, a drug that blocks the enzymatic activity of both Ezh1 and Ezh2 (Konze et al, 2013) tested the reactivation of several PRC/S5p and PRC Only genes by quantitative PCR. We find that PRC/S5p genes

become derepressed within 24h of treatment, in contrast with PRC Only genes, indicating that RNAPII occupancy increases the potential for reactivation when Polycomb inhibition is interfered with (Fig. 6C). These results are presented and discussed in page 19.

All together the results in the new Fig 6 support the concept that the presence of RNAPIIS5p makes PRC-repressed genes more susceptible to respond upon Polycomb knockout or chemical inhibition.

On the other hand, the authors should discuss their current and previous findings with respect to recent reports that challenge the poisoning of PRC genes by RNAPII (Williams et al, Mol Cell, 2015; Min et al., Genes Dev, 2011; Riising et al., Mol Cell, 2014). In these papers the authors report that most PRC genes in mESC display low or undetectable levels of poised RNAPII using GRO-seq or total RNAPII ChIP-seq. Moreover, they also showed that PRC2 is not necessary to maintain PRC-genes in a silent state in mESC.

Some of the discrepancies highlighted by the reviewer are likely to result from the use of different growth conditions for ESCs, especially the use of 2i. To clarify this point, we have expanded the introduction to explain that the poised RNAPII-S5p state found at PRC-repressed genes results from ERK1/2 phosphorylation (Ma et al, 2016; Tee et al, 2014), which is inhibited in 2i conditions. Furthermore, Williams et al (2015) and Riising et al. (2014) study ESCs grown in 2i condition, under conditions where ERK1/2 are inhibited, which is likely to account for the low levels of detection of RNAPII at Polycomb target genes. We have included the following text in page 5 and 6:
“The poised RNAPII-S5p state was initially observed at Polycomb-repressed genes in ESCs grown in the presence of serum and leukemia inhibitor factor (LIF; (Brookes et al, 2012; Stock et al, 2007). Other studies grow ESCs in 2i conditions to simulate a more naïve pluripotent state, through inhibition of GSK3 and MEK signalling, which in turn inhibits ERK signalling. In these conditions, the occupancy of poised RNAPII complexes is reduced at Polycomb-target genes (Marks et al, 2012; Williams et al, 2015), consistent with the effects of ERK1-2 inhibition (Tee et al, 2014). Interestingly, the decreased occupancy of poised RNAPII-S5p at PRC-repressed genes in 2i conditions is accompanied by reduced occupancy of PRC1 catalytic subunit EZH2 and H3K27me3 modification, suggesting a tight interplay between the presence of poised RNAPII-S5p and Polycomb occupancy at Polycomb-repressed genes in ESCs, which is interfered upon in 2i conditions. Interestingly, prolonged 2i treatment was shown to impair ESC developmental potential and cause widespread loss of DNA methylation (Choi et al, 2017; Yagi et al, 2017), leading to renewed interest in understanding the regulation of developmental genes, and in particular whether poised RNAPII complexes are a more general feature of Polycomb repression mechanisms in the early and late stages of differentiation.”

“Can the authors generate GRO-seq or total RNAPII ChIP-seq to test their findings based on S5P RNAPII ChIP-seq?. Alternatively, they could at least use publically available data in mESC.”

[Part of this answer has been removed, since it discusses unpublished data.]

... We hope that these unpublished results reassure the reviewers that the RNAPII-S5p complexes exist and have functional relevance for understanding mechanisms of Polycomb-repression and RNAPII regulation in mammalian cells. Many ill-understood aspects of the poised RNAPII-S5p complexes remain to be clarified in the future.

Until now, in part confounded by the immense confusion in nomenclature in different states of RNAPII activation, the remarkable poised RNAPII-S5p complexes at PRC-target genes had only been identified in ESCs in serum+LIF conditions. In our present study, we generalize this state and show it exists in all conditions investigated, from ESCs grown in serum-free conditions to terminally differentiated dopaminergic neurons. Showing that the poised S5p state is general in mammalian cells is a first step to help raise further interest towards deeper investigations of how poised RNAPII complexes are established and regulated, which genes are kept silenced through this mechanism, and how Polycomb repression

interplays with RNAPII regulation to establish the S5p poised state of activation.

2nd Editorial Decision

04 September 2017

Thank you again for sending us your revised manuscript. We have now heard back from Reviewer #2 who was asked to evaluate your study. As you will see below, s/he thinks that the reviewers' concerns have been satisfactorily addressed and the study is now suitable for publication. S/he only suggests some minor text modifications and recommends adding "Rebuttal Figure 2" as an Appendix Figure, since it is mainly based on published data.

REFEREE REPORT

Reviewer #2:

The authors have successfully and thoroughly addressed most of my concerns and, in my modest opinion, also of the other reviewer. I really appreciate that they shared with us some of their unpublished work to further support the presence of RNAPII-S5p complexes at a subset of PRC genes. In fact, given the confusion in the field, which the authors also acknowledge, I would encourage the authors to include in the current manuscript the Rebuttal Figure 2, as this figure is largely based on already published data. Moreover, such Figure could be complemented with a similar Boxplot panel using mRNA RNA-seq instead of GRO-seq data. Finally, I would also suggest the authors to use the term de-repressed rather than induced, re-activated or activated when referring to the gene expression changes observed upon loss of PRC inhibition. These changes are rather moderate in comparison to the expression levels that PRC target genes display once truly activated during embryogenesis and, thus, the term de-repression seems more appropriate in the context of their manuscript.

2nd Revision - authors' response

08 September 2017

We have addressed the minor requests from reviewer #2. However, the suggestion of Reviewer #2 to include our Rebuttal Fig 2 in the current manuscript is somewhat difficult to implement. This Rebuttal figure contains analyses of published datasets, which we have prepared in the context of a different manuscript. The difficulty in including Rebuttal Fig 2 is that the data is plotted from three specific lists of genes classified in a very specific manner based on published RNAPII and Polycomb datasets which we never use in the current paper. Either we would need to explain these groups (which would not make sense for the reader) in the paper and add all the respective methods or we would need to produce the graphs again for lists of genes that are directly relevant to our manuscript (which will give essentially the same results). We would therefore propose to not follow this recommendation, as it seems to be an optional point.

Corresponding Author Name: Ana Pombo
Manuscript Number: MSB-17-7754RR